# FEATURE LEARNING FOR THE HIGH DIMENSIONAL STATIONARY SCHÖDINGER EQUATION WITH DEEP RITZ METHOD

## ABSTRACT

This paper investigates feature learning within the framework of the deep Ritz method for solving the stationary Schrödinger equation with Neumann boundary conditions. We first analyze the convergence of Riemannian gradient descent in an agnostic setting, where the hypothesis function is restricted to a single-index model while the PDE solution is arbitrary. We prove that gradient descent reaches an approximate global minimum: after $T = O(\log(1/\epsilon))$ iterations, the loss is within $\epsilon$ of a constant multiple of the optimal loss. We then examine the loss landscape when the source term of the PDE itself follows a single-index model, considering hypothesis functions given by either a single-index model or a two-neuron multi-index model. In the single-index case, we show that the minimum Ritz energy is attained at the feature vector aligned with that of the source term. In the two-neuron case, we study the landscape of regularized Ritz losses and characterize how a second feature emerges, given that the first feature is aligned with the source, as the regularization parameter varies. Finally, numerical experiments are presented to validate the feature emergence theory in the two-neuron setting.

## 1 INTRODUCTION

The past decade has witnessed remarkable progress in applying deep learning techniques to scientific computing, and in particular to the numerical solution of partial differential equations (PDEs). Classical numerical methods, such as finite difference, finite element, and spectral methods, often face challenges when addressing high-dimensional problems, irregular domains, or when solutions exhibit complex structures. Neural network–based approaches have emerged as promising alternatives, offering mesh-free approximations, strong expressive power, and flexibility in incorporating physical constraints. Several notable frameworks have been proposed for solving PDEs with neural networks. The deep Ritz method (E & Yu, 2018) formulates the variational problem associated with elliptic PDEs as an energy minimization task and leverages neural networks as trial functions. Physics-informed neural networks (PINNs) (Raissi et al., 2019) instead enforce PDE constraints through the residual of the governing equations, incorporating them into the training loss. The Deep Galerkin Method (Sirignano & Spiliopoulos, 2018) generalizes this perspective by introducing stochastic collocation strategies to enforce weak PDE formulations. More recently, Weak Adversarial Networks (Zang et al., 2020) have explored adversarial training principles to approximate PDE solutions in a weak sense.

Despite these advances, the theoretical foundations of deep learning for PDEs lag behind their empirical success. While substantial progress has been made in understanding approximation power and generalization error, much less is known about **how neural networks learn effective features or representations of PDE solutions**. *Feature learning* — the process of automatically discovering meaningful low-dimensional structure from high-dimensional data, is a key ingredient to the success of neural networks in a broad range of machine learning tasks. Many recent theoretical works have demonstrated that neural networks trained with gradient-based algorithms are able to learn certain high dimensional target functions with low-dimensional structures, e.g. single- or multi-index models, in regression settings. However, to the best of our knowledge, the emergence of feature learning has not been systematically studied in the context of neural network–based methods for PDEs.

In this work, we investigate the feature learning mechanism of simple neural network models applied to high dimensional stationary Schödinger equation. Specifically, we consider fitting the solution of the Schödinger equation in the framework of deep Ritz method where the hypothesis function is defined by either a single index model or a two-neuron network. Specifically, we study solutions within the framework of the deep Ritz method, where the hypothesis function is either a single-index model or a two-neuron network. Our overarching goal is to characterize the feature directions that minimize the Ritz loss and to identify which of these directions can be effectively captured by gradient descent.

## 1.1 OUR CONTRIBUTIONS

We highlight the major contributions of the paper as follows:

- We first investigate the convergence guarantees of gradient descent (GD) in the agnostic setting, where the hypothesis function is restricted to a single-index model while the PDE solution is generic. We show that GD achieves an approximate global minimum in the sense that the loss value after $T = O(\log 1/\epsilon)$ iterations of GD is within $\epsilon$ of a constant multiplier of the minimum loss.

- Next, we focus on the loss landscape when the source term of the PDE is itself a single-index model. We consider two cases for the hypothesis function: a single-index model or a multi-index model with two neurons. In the first case, we prove that the minimum Ritz energy is attained at the same feature vector as the source term. In the latter case, we analyze the landscape of a regularized Ritz loss, where the regularization is applied to the outer-layer weights, in the high-dimensional regime. We characterize the sets of local and global minimizers of the high-dimensional limit of the Ritz loss as the regularization parameter varies. Our main results demonstrate that, when one feature vector of the hypothesis function aligns with that of the source term, the behavior of the second feature vector depends on the strength of the regularization: the regularized Ritz loss may admit a local or global minimizer that deviates from the feature vector of the source term.

- Finally, we provide numerical experiments to validate the theory we established for the loss landscape of the two-neuron multi-index model.

## 1.2 RELATED WORK

**Analysis of neural networks for PDEs.** Many recent theoretical studies have investigated the approximation power (Marwah et al., 2021; 2023; Grohs et al., 2022; 2023; Grohs & Herrmann, 2022; De Ryck & Mishra, 2024) of neural networks for representing solutions of PDEs. A number of works have also analyzed generalization error estimates (Mishra & Molinaro, 2023; De Ryck & Mishra, 2022; Shin et al., 2023; Lu et al., 2022b) within variational frameworks such as PINNs and the deep Ritz method. These analyses typically assume that PDE solutions lie in Sobolev or Hölder spaces, which leads to convergence rates that suffer from the curse of dimensionality: achieving an $\epsilon$-accurate approximation of a $d$-dimensional solution requires $n = O(\epsilon^{-cd})$ network parameters and training samples. In contrast, another line of research (Chen et al., 2021; 2023; Feng & Lu, 2025; Weinan & Wojtowytsch, 2022) has established dimension-free approximation rates for certain high-dimensional PDEs by developing new regularity theory in Barron spaces. More recently, dimension-free generalization error bounds (Lu et al., 2021a; Lu & Lu, 2022) have also been derived for specific classes of elliptic PDEs. Also, there is a series of stochastic differential equation (SDE)-based neural PDE solvers for elliptic equations (Nüsken & Richter, 2021; Han et al., 2020; Nam et al., 2024).

Compared to approximation-theoretic and generalization analyses, the optimization aspect of neural networks for PDEs remains significantly more challenging and less well understood, primarily due to the highly non-convex nature of the associated loss landscapes. Several recent works (Luo & Yang, 2024; Bonfanti et al., 2024; Zhao & Luo, 2025; Xu et al., 2024; Gao et al., 2023; Jiao et al., 2024) have investigated the convergence of gradient-based algorithms for training highly over-parameterized neural networks within the Neural Tangent Kernel (NTK) framework (Jacot et al., 2018). However, NTK-based analyses are intrinsically tied to the lazy training regime and thus fail to capture feature learning. An alternative line of research considers the mean-field regime (Mei et al., 2018; Sirignano & Spiliopoulos, 2020; Rotskoff & Vanden-Eijnden, 2018), where over-parameterized neural networks are studied via their distributional dynamics, described by Wasser-

stein gradient flows. In this setting, Dus & Virginie (2024); Dus & Ehrlacher (2025) established convergence results for Wasserstein gradient flows of the Ritz energy with infinite-width two-layer neural networks, applied to problems such as the Poisson equation and the Schrödinger eigenvalue problem. Nonetheless, these mean-field results are limited to the infinite-width setting and do not directly extend to finite-width networks, where feature learning and optimization dynamics remain far less understood.

**Feature learning of neural networks with gradient-based algorithms.** Beyond the NTK regime, a growing body of work seeks to understand how neural networks trained with gradient descent (GD) can recover low-dimensional structures (feature directions) of relatively simple target functions in regression problems. Examples include polynomials Yehudai & Shamir (2019); Damian et al. (2022), single-index models Soltanolkotabi (2017); Dudeja & Hsu (2018); Damian et al. (2023); Bietti et al. (2022), multi-index models Ba et al. (2022); Dandi et al. (2024); Cui et al. (2024); Moniri et al. (2024); Bruna & Hsu (2025), and sparse Boolean functions Abbe et al. (2022; 2023), among others. A central challenge in extending these analyses from regression to PDE problems lies in the intrinsic mismatch between the PDE solutions and the neural network approximators. Although PDE solutions may exhibit low-dimensional structures, they are typically far more complex than simple neural network models such as multi-index functions. This places the learning problem into the so-called *agnostic* setting, where the hypothesis class does not perfectly capture the target function. While a line of research investigated the first-order methods to learn agnostically the single-index Frei et al. (2020); Wu (2022); Awasthi et al. (2023); Wang et al. (2023); Gollakota et al. (2023); Zarifis et al. (2024) and multi-index models Diakonikolas et al. (2024), these results do not directly generalize to PDEs because the loss functions in PDE learning involve complex differential operators. A rigorous understanding of feature learning in neural network–based PDE solvers remains largely open, even for simple architectures.

### 1.3 NOTATION

We use bold uppercase and lowercase letters to denote matrices and column vectors, respectively. For $\boldsymbol{x} \in \mathbb{R}^d$, we denote its $p$-norm by $|\boldsymbol{x}|_p$. When $p = 2$, we write $|\boldsymbol{x}| = |\boldsymbol{x}|_2$. Given a function $f$, we use $\|f\|_\infty$ to denote the sup norm of $f$. Let $\Omega = \mathcal{B}^d$ be the unit ball on $\mathbb{R}^d$ and $\partial\Omega$ be the boundary of $\Omega$. Also, $\mathcal{S}^{d-1}$ denotes the sphere in $d$-dimension, $\mathcal{S}^{d-1} := \{\boldsymbol{x} \in \mathbb{R}^d : |\boldsymbol{x}| = 1\}$. We denote by $H^1(\Omega)$ the Sobolev space of square-integrable functions with square-integrable first derivatives. Given a unit vector $\boldsymbol{w} \in \mathcal{S}^{d-1}$, $\mathrm{P}_{\boldsymbol{w}^\perp} := I - \boldsymbol{w}\boldsymbol{w}^\top$ denotes the orthogonal projector onto the hyperplane perpendicular to $\boldsymbol{w}$, i.e., for $\boldsymbol{v} \in \mathbb{R}^d$, $\mathrm{P}_{\boldsymbol{w}^\perp}\boldsymbol{v} = \boldsymbol{v}^{\perp\boldsymbol{w}}$.

## 2 SET-UP AND MAIN RESULTS

We consider the following stationary Schrödinger equation

$$-\Delta u + u = f \text{ on } \Omega, \quad \frac{\partial u}{\partial \nu} = 0 \text{ on } \partial\Omega. \tag{1}$$

We assume that $f$ is known and $f \in L^2(\Omega)$. Thanks to the standard well-posedness of PDEs, there exists a unique weak solution $u^* \in H^1(\Omega)$ to the equation (1) and

$$u^* = \operatorname*{arg\,min}_{u \in H^1(\Omega)} \mathcal{E}(u) := \operatorname*{arg\,min}_{u \in H^1(\Omega)} \frac{1}{2} \int_\Omega |\nabla u|^2 + u^2 - 2fu\, dx. \tag{2}$$

It is also useful to note that for any $u \in H^1(\Omega)$,

$$\begin{aligned}
\mathcal{E}(u) &= \frac{1}{2} \int_\Omega |\nabla u|^2 + u^2 - 2(-\Delta u^* + u^*)u\, dx \\
&= \frac{1}{2} \int_\Omega |\nabla u - \nabla u^*|^2 + |u - u^*|^2 dx - \frac{1}{2} \int_\Omega |\nabla u^*|^2 + |u^*|^2 dx \\
&= \frac{1}{2} \int_\Omega |\nabla u - \nabla u^*|^2 + |u - u^*|^2 dx + \mathcal{E}(u^*),
\end{aligned}$$

where we have used integration by parts in the second equality. As a consequence, minimizing the energy $\mathcal{E}$ is equivalent to minimizing the shifted loss function $\mathcal{L}$ defined as follows:

$$
\begin{aligned}
\mathcal{L}(u) :=& \mathcal{E}(u) - \mathcal{E}(u^*) \\
=& \frac{1}{2} \int_\Omega |\nabla u - \nabla u^*|^2 + |u - u^*|^2 dx \\
=& \frac{|\Omega|}{2} \mathbb{E}_{x \sim \mathcal{P}_\Omega} \left[ |\nabla u(x) - \nabla u^*(x)|^2 + |u(x) - u^*(x)|^2 \right],
\end{aligned}
\tag{3}
$$

with $\mathcal{P}_\Omega$ denoting the uniform probability distributions on the domain $\Omega$.

The deep Ritz method (DRM) E & Yu (2018) seeks an approximate solution of (2) by minimizing the energy $\mathcal{E}$ (or its empirical version) within a hypothesis function class $\mathcal{F}$ which is parameterized by neural networks. The resulting optimization problem over the neural network parameters is non-convex in general and it remains a challenging open problem whether standard gradient-based algorithms can find the globally optimal solution. The purpose of this paper is to approach this problem by understanding the feature learning mechanism of gradient descent in the minimization of the DRM loss. Specifically, we perform the analysis in three settings as discussed in what follows.

## 2.1 SINGLE-INDEX HYPOTHESIS IN THE FULLY AGNOSTIC SETTING

We first consider the case where the hypothesis function $u$ is defined by a single-index model parameterized with a unit vector $\boldsymbol{w} \in \mathcal{S}^{d-1}$. More precisely, we assume that

$$
u(\boldsymbol{x}) \equiv u_{\boldsymbol{w}}(\boldsymbol{x}) = \sigma^2(\boldsymbol{w} \cdot \boldsymbol{x}), \boldsymbol{x} \in \Omega
\tag{4}
$$

where the activation function $\sigma(\cdot)$ is the ReLU function (i.e., $\sigma(\cdot) = \max\{0, \cdot\}$). We consider the squared ReLU activation instead of ReLU its own as it yields a differentiable (Lipschitz continuous) loss function (see equation (5)) leading to a well-defined gradient descent algorithm. Also, it has been shown that shallow neural networks with squared ReLU or high-order power of ReLU activation enjoy quantitative approximation rate in Sobolev spaces (Lu et al., 2022a; Mao et al., 2024). Moreover, we make the following a-priori assumption on the right hand side $f$ and the exact solution $u^*$. It is important to note that the ground-truth solution $u^*$ still belongs to a large function class and does not necessarily admit the same single-index form (4) as $u$. This places the problem in the fully *agnostic learning* setting.

**Assumption 2.1.** *The following statements hold for the equation (1).*

1. *There exist a constant $C_{u*}$ such that $\|\nabla u^*\|_\infty \leqslant C_{u*}$ and $\|u^*\|_\infty \leqslant C_{u*}$.*

2. *There exists a constant $C_f$ such that $\|f\|_\infty \leqslant C_f$.*

Ignoring the constant $\frac{|\Omega|}{2}$ and noting that $\sigma(\cdot)\sigma'(\cdot) = \sigma(\cdot)$ in the subgradient sense, one redefines the equivalent loss functions as follows

$$
\begin{aligned}
\mathcal{L}(\boldsymbol{w}) :=& \mathbb{E}_{\boldsymbol{x} \sim \mathcal{P}_\Omega} \left[ |\nabla_{\boldsymbol{x}} u_{\boldsymbol{w}}(\boldsymbol{x}) - \nabla_{\boldsymbol{x}} u^*(\boldsymbol{x})|^2 + |u_{\boldsymbol{w}}(\boldsymbol{x}) - u^*(\boldsymbol{x})|^2 \right] \\
=& \mathbb{E}_{\boldsymbol{x} \sim \mathcal{P}_\Omega} \left[ |2\sigma(\boldsymbol{w} \cdot \boldsymbol{x})\boldsymbol{w} - \nabla_{\boldsymbol{x}} u^*(\boldsymbol{x})|^2 + |\sigma^2(\boldsymbol{w} \cdot \boldsymbol{x}) - u^*(\boldsymbol{x})|^2 \right]
\end{aligned}
\tag{5}
$$

and

$$
\begin{aligned}
\mathcal{E}(\boldsymbol{w}) :=& \mathbb{E}_{\boldsymbol{x} \sim \mathcal{P}_\Omega} \left[ |\nabla_{\boldsymbol{x}} u_{\boldsymbol{w}}(\boldsymbol{x})|^2 + |u_{\boldsymbol{w}}(\boldsymbol{x})|^2 - 2f(\boldsymbol{x})u_{\boldsymbol{w}}(\boldsymbol{x}) \right] \\
=& \mathbb{E}_{\boldsymbol{x} \sim \mathcal{P}_\Omega} \left[ |2\sigma(\boldsymbol{w} \cdot \boldsymbol{x})\boldsymbol{w}|^2 + |\sigma^2(\boldsymbol{w} \cdot \boldsymbol{x})|^2 - 2f(\boldsymbol{x})\sigma^2(\boldsymbol{w} \cdot \boldsymbol{x}) \right].
\end{aligned}
$$

Moreover, since $\mathcal{L}(\boldsymbol{w})$ and $\mathcal{E}(\boldsymbol{w})$ only differs by a constant (see 3), one has that $\nabla \mathcal{L}(\boldsymbol{w}) = \nabla \mathcal{E}(\boldsymbol{w})$.

### 2.1.1 OPTIMIZATION VIA RIEMANNIAN GRADIENT DESCENT

To optimize the loss $\mathcal{L}$ defined by 5, we adopt the classic Riemannian gradient descent (GD) method. First, observe that the gradient of the loss function $\mathcal{L}(\boldsymbol{w})$ with respect to $\boldsymbol{w}$ is given by

$$
\begin{aligned}
\nabla \mathcal{L}(\boldsymbol{w}) =& \mathbb{E}_{\boldsymbol{x} \sim \mathcal{P}_\Omega} \left[ 4\sigma'(\boldsymbol{w} \cdot \boldsymbol{x})\boldsymbol{x}\boldsymbol{w}^\top (2\sigma(\boldsymbol{w} \cdot \boldsymbol{x})\boldsymbol{w} - \nabla u^*) + 4(2\sigma(\boldsymbol{w} \cdot \boldsymbol{x})\boldsymbol{w} - \nabla u^*)\sigma(\boldsymbol{w} \cdot \boldsymbol{x}) \right] \\
& + \mathbb{E}_{\boldsymbol{x} \sim \mathcal{P}_\Omega} \left[ 4(\sigma^2(\boldsymbol{w} \cdot \boldsymbol{x}) - u^*(\boldsymbol{x}))\sigma(\boldsymbol{w} \cdot \boldsymbol{x})\boldsymbol{x} \right].
\end{aligned}
$$

However, the expression above is intractable since the ground-truth $u^*$ is unknown. Instead, noting that $\nabla \mathcal{L}(\boldsymbol{w}) = \nabla \mathcal{E}(\boldsymbol{w})$, we use the tractable gradient $\nabla \mathcal{E}(\boldsymbol{w})$ of the Ritz energy $\mathcal{E}$ given by

$$\nabla \mathcal{E}(\boldsymbol{w}) = 4\mathbb{E}_{\boldsymbol{x} \sim \mathcal{P}_\Omega} \left[ 2|\boldsymbol{w}|^2 \sigma(\boldsymbol{w} \cdot \boldsymbol{x})\boldsymbol{x} + 2\sigma^2(\boldsymbol{w} \cdot \boldsymbol{x})\boldsymbol{w} + \sigma^3(\boldsymbol{w} \cdot \boldsymbol{x})\boldsymbol{x} - f(\boldsymbol{x})\sigma(\boldsymbol{w} \cdot \boldsymbol{x})\boldsymbol{x} \right].$$

Recall that $\mathrm{P}_{\boldsymbol{w}^\perp} := \boldsymbol{I} - \boldsymbol{w}\boldsymbol{w}^\top$. Then the Riemannian gradients of the loss function $\mathcal{L}(\boldsymbol{w})$ and the energy function $\mathcal{E}(\boldsymbol{w})$, denoted by $g_\mathcal{L}(\boldsymbol{w})$ and $g_\mathcal{E}(\boldsymbol{w})$ respectively are $g_\mathcal{L}(\boldsymbol{w}) := \mathrm{P}_{\boldsymbol{w}^\perp}\nabla\mathcal{L}(\boldsymbol{w})$ and $g_\mathcal{E}(\boldsymbol{w}) := \mathrm{P}_{\boldsymbol{w}^\perp}\nabla\mathcal{E}(\boldsymbol{w})$. Specifically, the gradient of the Ritz energy can be evaluated explicitly as

$$g_\mathcal{E}(\boldsymbol{w}) = 4\mathbb{E}_{\boldsymbol{x} \sim \mathcal{P}_\Omega} \left[ 2|\boldsymbol{w}|^2 \sigma(\boldsymbol{w} \cdot \boldsymbol{x})\mathrm{P}_{\boldsymbol{w}^\perp}\boldsymbol{x} + \sigma^3(\boldsymbol{w} \cdot \boldsymbol{x})\mathrm{P}_{\boldsymbol{w}^\perp}\boldsymbol{x} - f(\boldsymbol{x})\sigma(\boldsymbol{w} \cdot \boldsymbol{x})\mathrm{P}_{\boldsymbol{w}^\perp}\boldsymbol{x} \right].$$

In practice, the expectation above is approximated by an empirical estimator $\hat{g}_\mathcal{E}$ computed with $n$ iid uniform samples $\{\boldsymbol{x}_i\}_{i=1}^n$ on $\Omega$ by

$$\hat{g}_\mathcal{E}(\boldsymbol{w}^t) := \frac{1}{n}\sum_{i=1}^n 4\left( 2|\boldsymbol{w}^t|^2 \sigma(\boldsymbol{w}^t \cdot \boldsymbol{x}_i)\mathrm{P}_{(\boldsymbol{w}^t)^\perp}\boldsymbol{x}_i + \sigma^3(\boldsymbol{w}^t \cdot \boldsymbol{x}_i)\mathrm{P}_{(\boldsymbol{w}^t)^\perp}\boldsymbol{x}_i - f(\boldsymbol{x}_i)\sigma(\boldsymbol{w}^t \cdot \boldsymbol{x}_i)\mathrm{P}_{(\boldsymbol{w}^t)^\perp}\boldsymbol{x}_i \right).$$

We summarize the gradient descent procedure in Algorithm 1. It takes as input an initial guess $\boldsymbol{w}^0$ for the desired parameter, accuracy tolerance parameter $\epsilon$ (see Theorem 2.2), number of iterations $T$, step size $\eta$, and the sample distribution $\mathcal{P}_\Omega$. It outputs the estimated parameter $\boldsymbol{w}^T$ obtained by the Riemannian GD.

---

**Algorithm 1** Riemannian GD

---

1: **Input:** $\boldsymbol{w}^0$, $\epsilon$, $T$, $\eta$; Sample distribution $\mathcal{P}_\Omega$.
2: Draw $n = \Theta(d(d+2)/\epsilon)$ samples $\{\boldsymbol{x}_i\}_{i=1}^n$ from $\mathcal{P}_\Omega$.
3: **for** $t = 0, \cdots, T-1$ **do**
4:     Compute the empirical estimate of $g_\mathcal{E}(\boldsymbol{w}^t)$:

$$\hat{g}_\mathcal{E}(\boldsymbol{w}^t) := \frac{1}{n}\sum_{i=1}^n 4\left( 2|\boldsymbol{w}^t|^2 \sigma(\boldsymbol{w}^t \cdot \boldsymbol{x}_i)\mathrm{P}_{(\boldsymbol{w}^t)^\perp}\boldsymbol{x}_i + \sigma^3(\boldsymbol{w}^t \cdot \boldsymbol{x}_i)\mathrm{P}_{(\boldsymbol{w}^t)^\perp}\boldsymbol{x}_i - f(\boldsymbol{x}_i)\sigma(\boldsymbol{w}^t \cdot \boldsymbol{x}_i)\mathrm{P}_{(\boldsymbol{w}^t)^\perp}\boldsymbol{x}_i \right).$$

5:     Gradient descent and normalize: $\boldsymbol{w}^{t+1} = (\boldsymbol{w}^t - \eta\hat{g}_\mathcal{E}(\boldsymbol{w}^t))/|\boldsymbol{w}^t - \eta\hat{g}_\mathcal{E}(\boldsymbol{w}^t)|_2$.
   **end for**
6: **return** $\boldsymbol{w}^T$

---

We measure the performance of $\boldsymbol{w}^T$ by comparing the loss value $\mathcal{L}(\boldsymbol{w}^T)$ with the minimum loss value OPT defined by

$$\mathrm{OPT} := \mathcal{L}(\boldsymbol{w}^*) = \mathbb{E}_{\boldsymbol{x} \sim \mathcal{P}_\Omega}\left[ |\nabla_{\boldsymbol{x}} u_{\boldsymbol{w}^*}(\boldsymbol{x}) - \nabla_{\boldsymbol{x}} u^*(\boldsymbol{x})|^2 + |u_{w^*}(\boldsymbol{x}) - u^*(\boldsymbol{x})|^2 \right],$$

where $\boldsymbol{w}^* := \arg\min_{\boldsymbol{w} \in \mathcal{S}^{d-1}} \mathcal{L}(\boldsymbol{w})$. Our first main theorem below shows that Algorithm 1 produces an approximate solution $u_{\boldsymbol{w}^T}$ to the problem (5) after $T = O(\log(1/\epsilon))$ iterations in the sense that the loss value $\mathcal{L}(\boldsymbol{w}^T)$ is within $\epsilon$ of a constant multiplier of the OPT.

**Theorem 2.2.** *Suppose that Assumption 2.1 holds. Consider Algorithm 1 with initial condition $\boldsymbol{w}^0$ satisfying $\angle(\boldsymbol{w}^0, \boldsymbol{w}^*) \in [0, \frac{\pi}{2}]$. Set the sample size $n = \Theta\left(\frac{d(d+2)}{\epsilon}\right)$ and the step size $\eta = \frac{d+2}{2048\pi}$. Then after $T = O(\log(1/\epsilon))$ iterations, with probability at least $1 - \delta$, the output of Algorithm 1, $\boldsymbol{w}^T$, satisfies $\mathcal{L}(\boldsymbol{w}^T) < \gamma \cdot \mathrm{OPT} + \frac{128}{d+2}\epsilon$, where the absolute constant $0 < \gamma < 2048\pi^2 + 2$.*

The proof of Theorem 2.2 can be found in Appendix A. We remark that the linear dependence of the step size $\eta$ on the dimension $d$ arises from the fact that the Riemannian gradient satisfies $g_\mathcal{E}(\boldsymbol{w}) = O(1/d)$. Consequently, scaling the step size with $d$ ensures that the effective update of magnitude remains balanced and prevents vanishingly small progress in high dimensions.

## 2.2 SINGLE-INDEX HYPOTHESIS

In this subsection, we focus on the setting where $f$ is given by a single-index model $f(\boldsymbol{x}) = \sigma^2(\boldsymbol{w}^* \cdot \boldsymbol{x})$ for some $\boldsymbol{w}^* \in \mathcal{S}^{d-1}$. We also consider the hypothesis function $u$ is defined by a single index model $u(\boldsymbol{x}) = \sigma^2(\boldsymbol{w} \cdot \boldsymbol{x})$. Then the problem of minimizing the Ritz energy becomes

$$\min_{\boldsymbol{w} \in \mathcal{S}^{d-1}} \mathcal{E}(\boldsymbol{w}) := \frac{|\Omega|}{2}\mathbb{E}_{\boldsymbol{x} \sim \mathcal{P}_\Omega}\left[ |2\sigma(\boldsymbol{w} \cdot \boldsymbol{x})\boldsymbol{w}|^2 + |\sigma^2(\boldsymbol{w} \cdot \boldsymbol{x})|^2 - 2\sigma^2(\boldsymbol{w}^* \cdot \boldsymbol{x})\sigma^2(\boldsymbol{w} \cdot \boldsymbol{x}) \right]. \quad (6)$$

Similar to the previous section, we remark that the exact solution $u^*$ associated to the source term $f$ is not a single index model. However, the proposition below shows that the minimum of the Ritz energy function $\mathcal{E}(\boldsymbol{w})$, defined in (6), attains its minimum in the same direction $w^*$ as the source term $f$. We defer the proof of Proposition 2.3 to Appendix B.

**Proposition 2.3.** *The minimum value of the Ritz energy function $\mathcal{E}(\boldsymbol{w})$ is achieved when $\boldsymbol{w} = \boldsymbol{w}^*$.*

### 2.3 MULTI-INDEX HYPOTHESIS

In this subsection, we assume that $f$ is given by a single-index model $f(\boldsymbol{x}) = \sigma^2(\boldsymbol{w}^* \cdot \boldsymbol{x})$, but consider a more complicated hypothesis model — two-neuron neural network, which is a special case of *multi-index model*. More concretely, we assume that

$$u(\boldsymbol{x}) = \sum_{i=1}^{2} a_i \sigma^2(\boldsymbol{w}_i \cdot \boldsymbol{x})$$

with $\boldsymbol{w}_1, \boldsymbol{w}_2 \in \mathcal{S}^{d-1}$ and $a_1, a_2 \in \mathbb{R}$. Motivated by Proposition 2.3 for the single-index hypothesis function, we fix the first feature vector $\boldsymbol{w}_1 = \boldsymbol{w}^*$. In fact, we can also show that if the hypothesis function is $a_1\sigma^2(w_1 \cdot x) + a_2\sigma^2(w_2 \cdot x)$ with $a_i > 0$, then one of the features also aligns with $\boldsymbol{w}^*$. For simplicity, we investigate the mechanism by which the second feature $\boldsymbol{w}_2$ emerges under this constraint. Under this setting, the variational problem of minimizing the Ritz energy becomes

$$\mathcal{L}(\boldsymbol{w}, \boldsymbol{a}) := \mathbb{E}_{\boldsymbol{x} \sim \mathcal{P}_\Omega} \left[ |\nabla_{\boldsymbol{x}} u|^2 + |u|^2 - 2fu \right]$$

$$= \mathbb{E}_{\boldsymbol{x} \sim \mathcal{P}_\Omega} \left[ \left| 2 \sum_{i=1}^{2} a_i \sigma(\boldsymbol{w}_i \cdot x) \boldsymbol{w}_i \right|^2 + \left| \sum_{i=1}^{2} a_i \sigma^2(\boldsymbol{w}_i \cdot \boldsymbol{x}) \right|^2 - 2\sigma^2(\boldsymbol{w}^* \cdot \boldsymbol{x}) \sum_{i=1}^{2} a_i \sigma^2(\boldsymbol{w}_i \cdot \boldsymbol{x}) \right].$$
(7)

To write the loss function in a more compact form, we apply the similar techniques as in Cho & Saul (2009) and define the following kernels

$$K_1(\boldsymbol{w}_i, \boldsymbol{w}_j) := 4\boldsymbol{w}_i^\top \boldsymbol{w}_j \mathbb{E}_{\boldsymbol{x} \sim \mathcal{P}_\Omega}[\sigma(\boldsymbol{w}_i \cdot \boldsymbol{x})\sigma(\boldsymbol{w}_j \cdot \boldsymbol{x})] = \frac{2}{\pi(d+2)} h_1(\theta_{ij}),$$

$$K_2(\boldsymbol{w}_i, \boldsymbol{w}_j) := \mathbb{E}_{\boldsymbol{x} \sim \mathcal{P}_\Omega}[\sigma^2(\boldsymbol{w}_i \cdot \boldsymbol{x})\sigma^2(\boldsymbol{w}_j \cdot \boldsymbol{x})] = \frac{1}{2\pi(d+2)(d+4)} h_2(\theta_{ij}),$$

where $\theta_{ij} = \angle(\boldsymbol{w}_i, \boldsymbol{w}_j)$ is the angle between $w_i$ and $w_j$ and

$$h_1(\theta_{ij}) := (\sin\theta_{ij}\cos\theta_{ij} + (\pi - \theta_{ij})\cos^2\theta_{ij}),$$

$$h_2(\theta_{ij}) := 3\sin\theta_{ij}\cos\theta_{ij} + (\pi - \theta_{ij})(1 + 2\cos^2\theta_{ij}),$$

Note that we have also used $\boldsymbol{w}_i^\top \boldsymbol{w}_j = \cos\theta_{ij}$. Therefore, the loss function (7) can be written as

$$\mathcal{L}(\boldsymbol{w}, \boldsymbol{a}) = \boldsymbol{a}^\top \boldsymbol{K}_1 \boldsymbol{a} + \boldsymbol{a}^\top \boldsymbol{K}_2 \boldsymbol{a} - 2\boldsymbol{a}^\top \boldsymbol{K}_*,$$
(8)

where $\boldsymbol{K}_1 \in \mathbb{R}^{2 \times 2}$, $\boldsymbol{K}_2 \in \mathbb{R}^{2 \times 2}$ and $\boldsymbol{K}_* \in \mathbb{R}^{2 \times 1}$ are defined as follows:

$$\boldsymbol{K}_1 = \begin{bmatrix} K_1(\boldsymbol{w}_1, \boldsymbol{w}_1) & K_1(\boldsymbol{w}_1, \boldsymbol{w}_2) \\ K_1(\boldsymbol{w}_2, \boldsymbol{w}_1) & K_1(\boldsymbol{w}_2, \boldsymbol{w}_2) \end{bmatrix}, \boldsymbol{K}_2 = \begin{bmatrix} K_2(\boldsymbol{w}_1, \boldsymbol{w}_1) & K_2(\boldsymbol{w}_1, \boldsymbol{w}_2) \\ K_2(\boldsymbol{w}_2, \boldsymbol{w}_1) & K_2(\boldsymbol{w}_2, \boldsymbol{w}_2) \end{bmatrix}, \boldsymbol{K}_* = \begin{bmatrix} K_2(\boldsymbol{w}^*, \boldsymbol{w}_1) \\ K_2(\boldsymbol{w}^*, \boldsymbol{w}_2) \end{bmatrix}.$$

In practice, an $\ell_2$-regularization is commonly employed to promote weight decay. Here we consider two variants: one applied to $\boldsymbol{a} = (a_1, a_2)$ and another applied solely to $a_2$. In both regularization settings, our primary interest is to understand how the other feature $\boldsymbol{w}_2$ emerges in the high dimensional regime $d \to \infty$ given that the first feature is fixed at $\boldsymbol{w}_1 = \boldsymbol{w}^*$ as the regularization parameter $\lambda$ varies.

#### 2.3.1 REGULARIZATION APPLIED TO $\boldsymbol{a}$

We first consider the following regularized loss function:

$$\mathcal{L}_\lambda(\boldsymbol{w}, \boldsymbol{a}) = \boldsymbol{a}^\top \boldsymbol{K}_1 \boldsymbol{a} + \boldsymbol{a}^\top \boldsymbol{K}_2 \boldsymbol{a} - 2\boldsymbol{a}^\top \boldsymbol{K}_* + \lambda \boldsymbol{a}^\top \boldsymbol{a},$$
(9)

where $\lambda > 0$ is the regularization parameter. To minimize the regularized loss function with respect to the parameters $\boldsymbol{w}$ and $\boldsymbol{a}$, noting first that for any fixed $\boldsymbol{w}$, the function $\boldsymbol{a} \mapsto \mathcal{L}_\lambda(\boldsymbol{w}, \boldsymbol{a})$ is quadratic and the minimizer $\boldsymbol{a}^*$ has a closed form solution and the regularized loss function can thus be reduced to a loss function depending only on $\boldsymbol{w}$. The details are presented in the following lemma.

**Lemma 2.4.** *For any fixed $\boldsymbol{w}$, the minimizer $\boldsymbol{a}^*$ of the regularized loss function (9) has a closed form $\boldsymbol{a}^* = (\boldsymbol{K}_1 + \boldsymbol{K}_2 + \lambda \boldsymbol{I}_2)^{-1} \boldsymbol{K}_*$ and $\mathcal{L}_\lambda(\boldsymbol{w}) = -\boldsymbol{K}_*^\top (\boldsymbol{K}_1 + \boldsymbol{K}_2 + \lambda \boldsymbol{I}_2)^{-1} \boldsymbol{K}_*$.*

Since by assumption $\boldsymbol{w}_1$ is aligned with $\boldsymbol{w}^*$, the loss function $\mathcal{L}_\lambda(\boldsymbol{w})$ can be rewritten as a loss of the angle $\theta := \angle(\boldsymbol{w}^*, \boldsymbol{w}_2) \in [0, \pi]$. More precisely, letting $c = \frac{2}{d+2}$ and defining $\lambda = \xi c$ with a rescaled regularization parameter $\xi$, it can be shown that the loss $\mathcal{L}_\lambda(\boldsymbol{w}) = \mathcal{L}_\xi(\theta)$ where

$$\mathcal{L}_\xi(\theta)$$

$$= -\frac{c^2}{16\pi^2(d+4)^2} \begin{bmatrix} h_2(0) \\ h_2(\theta) \end{bmatrix}^\top \begin{bmatrix} \frac{c}{\pi} h_1(0) + \frac{c}{4\pi(d+4)} h_2(0) + \xi c & \frac{c}{\pi} h_1(\theta) + \frac{c}{4\pi(d+4)} h_2(\theta) \\ \frac{c}{\pi} h_1(\theta) + \frac{c}{4\pi(d+4)} h_2(\theta) & \frac{c}{\pi} h_1(0) + \frac{c}{4\pi(d+4)} h_2(0) + \xi c \end{bmatrix}^{-1} \begin{bmatrix} h_2(0) \\ h_2(\theta) \end{bmatrix}.$$

Since our focus is on feature learning in the high-dimensional regime and noting that $h_1(0) = \pi$ and $h_2(0) = 3\pi$, we define the following limiting function as $d \to +\infty$:

$$\widetilde{\mathcal{L}}_\xi(\theta) := \lim_{d \to +\infty} \frac{16(d+4)^2}{c} \mathcal{L}_\xi(\theta)$$

$$= -\frac{(9 + 9\xi) + \frac{1}{\pi^2}[(1+\xi)h_2^2(\theta) - 6h_1(\theta)h_2(\theta)]}{(1+\xi)^2 - \frac{1}{\pi^2}h_1^2(\theta)}. \tag{10}$$

The theorem below characterizes the set of minimizers of the limiting function $\widetilde{\mathcal{L}}_\xi(\theta)$ as $\xi$ varies.

**Theorem 2.5.** *Consider the minimization of the limiting loss function $\widetilde{\mathcal{L}}_\xi$ defined by* (10).

1. *When $\xi \geqslant \frac{1}{2}$, $\widetilde{\mathcal{L}}_\xi(\theta)$ has a unique global minimizer at $\theta = 0$ for $\theta$ on $[0, \frac{5\pi}{6}]$.*

2. *When $\xi \leqslant \xi_0$ with some $\xi_0 < 1/2$, besides the local minimizer $\theta = 0$, there exists at least one additional local minimizer of $\widetilde{\mathcal{L}}_\xi(\theta)$ in the interval $(\frac{\pi}{4}, \frac{\pi}{2})$.*

We defer the proofs of Lemma 2.4 and Theorem 2.5 to Appendix C.

**Remark**. Although we can only rigorously prove the above theorem, our numerical experiments indicate a phase transition between a local minimizer (unique global minimizer) and two local minimizers, one at 0 and the other lying in $(\pi/4, \pi/2)$. This phase transition occurs approximately at the value $\xi_0 = 0.13$. More details can be found in Figure 1.

### 2.3.2 Regularization applied to $a_2$ solely

We now consider the regularized loss function in which only the coefficient $a_2$ is penalized:

$$\mathcal{L}_\lambda(\boldsymbol{w}, \boldsymbol{a}) = \boldsymbol{a}^\top \boldsymbol{K}_1 \boldsymbol{a} + \boldsymbol{a}^\top \boldsymbol{K}_2 \boldsymbol{a} - 2\boldsymbol{a}^\top \boldsymbol{K}_* + \lambda a_2^2. \tag{11}$$

To minimize the above regularized loss function with respect to parameters $\boldsymbol{w}$ and $\boldsymbol{a}$, we first note that for any fixed $\boldsymbol{w}$, the minimizer $\boldsymbol{a}^*$ has a closed form solution and the regularized loss function can be converted to a loss function that is only about $\boldsymbol{w}$. The details are presented in the following lemma.

**Lemma 2.6.** *For any fixed $\boldsymbol{w}$, the minimizer $\boldsymbol{a}^*$ of the regularized loss function (11) has a closed form $\boldsymbol{a}^* = \left(\boldsymbol{K}_1 + \boldsymbol{K}_2 + \begin{bmatrix} 0 & 0 \\ 0 & \lambda \end{bmatrix}\right)^{-1} \boldsymbol{K}_*$ and $\mathcal{L}_\lambda(\boldsymbol{w}) = -\boldsymbol{K}_*^\top \left(\boldsymbol{K}_1 + \boldsymbol{K}_2 + \begin{bmatrix} 0 & 0 \\ 0 & \lambda \end{bmatrix}\right)^{-1} \boldsymbol{K}_*$.*

Given that $\boldsymbol{w}_1$ is aligned with $\boldsymbol{w}^*$, the loss function $\mathcal{L}_\lambda(\boldsymbol{w})$ is equivalent to the following loss function $\mathcal{L}_\xi(\theta)$, upon substituting the definitions of $\boldsymbol{K}_1$ and $\boldsymbol{K}_2$, where $\theta := \angle(\boldsymbol{w}^*, \boldsymbol{w}_2)$:

$$\mathcal{L}_\xi(\theta)$$

$$= -\frac{c^2}{16\pi^2(d+4)^2} \begin{bmatrix} h_2(0) \\ h_2(\theta) \end{bmatrix}^\top \begin{bmatrix} \frac{c}{\pi} h_1(0) + \frac{c}{4\pi(d+4)} h_2(0) & \frac{c}{\pi} h_1(\theta) + \frac{c}{4\pi(d+4)} h_2(\theta) \\ \frac{c}{\pi} h_1(\theta) + \frac{c}{4\pi(d+4)} h_2(\theta) & \frac{c}{\pi} h_1(0) + \frac{c}{4\pi(d+4)} h_2(0) + \xi c \end{bmatrix}^{-1} \begin{bmatrix} h_2(0) \\ h_2(\theta) \end{bmatrix},$$

where $c = \frac{2}{d+2}$ and $\lambda = \xi c$ for a reparameterized regularization parameter $\xi$. Similar to the previous section, we consider the following limiting function (as $d \to +\infty$):

$$\bar{\mathcal{L}}_\xi(\theta) := \lim_{d \to +\infty} \frac{16(d+4)^2}{c} \mathcal{L}_\xi(\theta)$$

$$= -\frac{(9 + 9\xi) + \frac{1}{\pi^2}[h_2^2(\theta) - 6h_1(\theta)h_2(\theta)]}{1 + \xi - \frac{1}{\pi^2}h_1^2(\theta)}. \tag{12}$$

We are now ready to present the main result of this subsection and defer the proof of Lemma 2.6 and Theorem 2.7 to Appendix D.

**Theorem 2.7.** *Consider the minimization of the limiting loss function $\bar{\mathcal{L}}_\xi$ defined by (12). For any $\xi > 0$, the function $\bar{\mathcal{L}}_\xi(\theta)$ has a **unique** global minimizer $\theta^* \in (\frac{\pi}{3}, \frac{\pi}{2})$.*

Comparing Theorem 2.7 with Theorem 2.5, we observe that different forms of regularization on $\boldsymbol{a}$ lead to distinct behaviors in feature emergence. In particular, penalizing only $a_2$ consistently yields the emergence of an additional feature, whereas strong penalization on both outer-layer weights results in feature collapse.

## 3 NUMERICAL EXPERIMENTS

In this section, we provide numerical results to validate our theory established in Section 2.3 for a multi-index model with two neurons.

First, we show the limiting function $\widetilde{\mathcal{L}}_\xi(\theta)$ in Figure 1 that corresponds to the regularized loss function with regularization applied on the whole vector $\boldsymbol{a}$ (Subsection 2.3.1). The left plot of Figure 1 shows the 3D surface defined by the function $(\xi, \theta) \mapsto \widetilde{\mathcal{L}}_\xi(\theta)$ with $\xi \in [0, 0.2]$ and $\theta \in [0, \pi]$. The right plot of the same figure shows the function $\widetilde{\mathcal{L}}_\xi(\theta)$ with some specifically chosen values of $\xi$. We also mark the global minimizers of $\widetilde{\mathcal{L}}_\xi(\theta)$ for $\xi = 0.01$, 0.08, 0.09. From the figure, we can observe the landscape of $\widetilde{\mathcal{L}}_\xi(\theta)$ exhibits the following phase transitions as $\xi$ varies:

- When $0 < \xi \leqslant \xi_0 \approx 0.08$, $\widetilde{\mathcal{L}}_\xi(\theta)$ has two local minimizers, one at $\theta = 0$ and the other is in the interval $(\frac{\pi}{3}, \frac{\pi}{2})$. Moreover, the global minimizer is in the interval $(\frac{\pi}{3}, \frac{\pi}{2})$.

- When $\xi_0 \approx 0.08 < \xi \leqslant \xi_1 \approx 0.13$, $\widetilde{\mathcal{L}}_\xi(\theta)$ has two local minimizers, one at $\theta = 0$ and the other is in the interval $(\frac{\pi}{4}, \frac{\pi}{2})$. Moreover, $\theta = 0$ is the global minimizer.

- When $\xi > \xi_1 \approx 0.13$, $\widetilde{\mathcal{L}}_\xi(\theta)$ has only one local (and hence global) minimizer at $\theta = 0$.

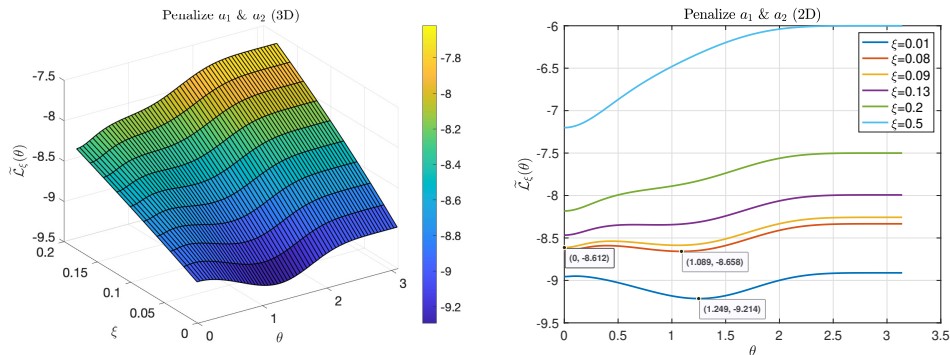

Figure 1: Graphs of the limiting function $\widetilde{\mathcal{L}}_\xi(\theta)$.

Next, we plot the limiting function $\bar{\mathcal{L}}_\xi(\theta)$ in Figure 2, which corresponds to the limiting regularized loss function with regularization applied only to $a_2$ defined by (12). In Figure 2, the left graph shows the 3D plot of the function $(\xi, \theta) \mapsto \bar{\mathcal{L}}_\xi(\theta)$ with $\xi \in [0.01, 0.5]$ and $\theta \in [0, \pi]$. The right graph shows plots of the function $\theta \mapsto \bar{\mathcal{L}}_\xi(\theta)$ with $\xi$ taking a broader range of values. We also mark the global minimizers of $\bar{\mathcal{L}}_\xi(\theta)$ for $\xi = 0.01$, 1, 5, 100. Observe that the loss function $\bar{\mathcal{L}}_\xi(\theta)$ has a unique global minimizer in the interval $(\frac{\pi}{3}, \frac{\pi}{2})$ for $\xi > 0$.

Moreover, we plot the loss function $\mathcal{L}_\xi(\theta)$ with $d = 2$ when the regularization is applied to $\boldsymbol{a}$ and $a_2$ solely in Figure 3. We observe that the loss function has the same shape as the corresponding limiting functions.

We also perform numerical experiments on the landscapes for sigmoid and GELU activation functions with $d = 2$ and observe that they have different angle landscapes (Figure 4). For sigmoid

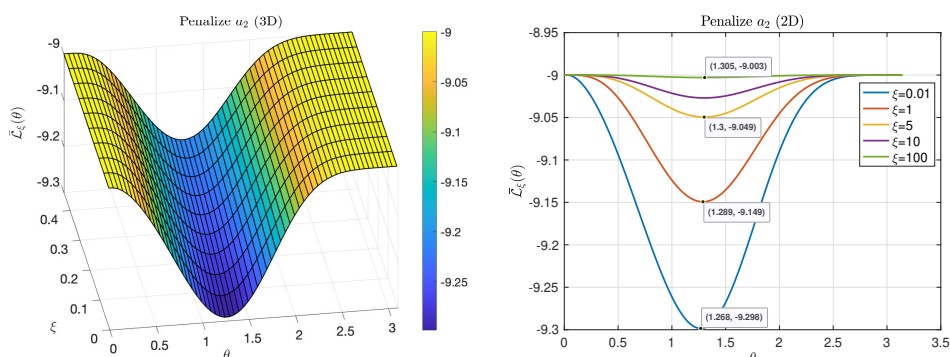

Figure 2: Graphs of the limiting function $\bar{\mathcal{L}}_\xi(\theta)$.

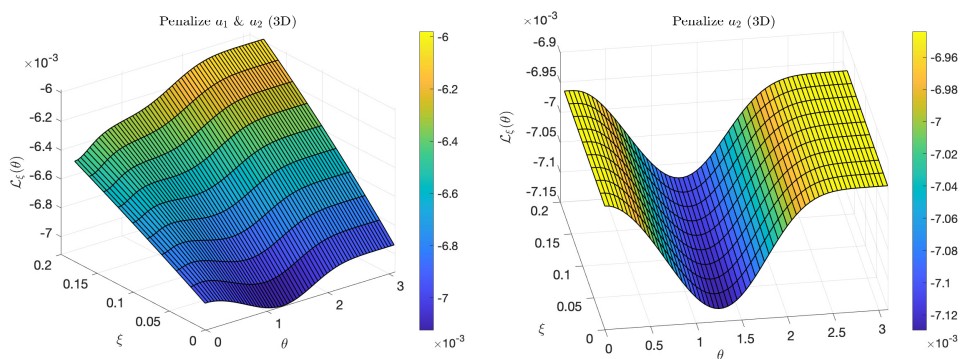

Figure 3: Graphs of the loss function $\mathcal{L}_\xi(\theta)$ with $d = 2$.

activation, the unique global minimum is achieved at $\theta = \pi$ when the regularization is applied to $\boldsymbol{a}$ and $a_2$ solely. For GELU, when the regularization is applied to $\boldsymbol{a}$, there are two local minimizers at $\theta = 0$ and $\theta = \pi$ and the global minimizer changes from $\theta = \pi$ to $\theta = 0$ as $\lambda$ increases. When the regularization is applied to $a_2$ solely, there are two local minimizers, one is in $(1, 1.5)$ and the other is at $\theta = \pi$ and the global minimizer is at $\theta = \pi$.

## 4 CONCLUSION AND DISCUSSION

In this work, we present a systematic study of feature learning in the context of stationary Schrödinger equation using the deep Ritz method. Unlike prior analyses that rely on infinite-width limits or strong over-parameterization assumptions, our study focuses on the behavior of finite-width models and examines how low-dimensional features of PDE solutions emerge as a result of an optimization procedure. Specifically, in an agnostic setting where the PDE solution is generic, we established convergence guarantees for gradient descent under a single-index hypothesis class, showing that approximate global optimality can be achieved in logarithmic iteration complexity. Further, when the source term of the PDE follows a single-index structure, we characterize the loss landscape for both single- and two-neuron models. Our analysis of the regularized Ritz losses revealed how feature emergence depends critically on the regularization strength: penalizing a single outer-layer weight consistently produces an additional feature, whereas strong joint penalization can result in feature collapse. Collectively, these results provide a first step toward a rigorous theory of feature learning in finite-width neural networks for PDEs.

Our work opens several directions for future research. While our analysis focused on single-index and two-neuron models, it remains an open question to investigate multi-index models with more neurons and to determine whether feature emergence exhibits hierarchical or sequential patterns in such settings. Another important direction is to extend the analysis to broader classes of PDEs

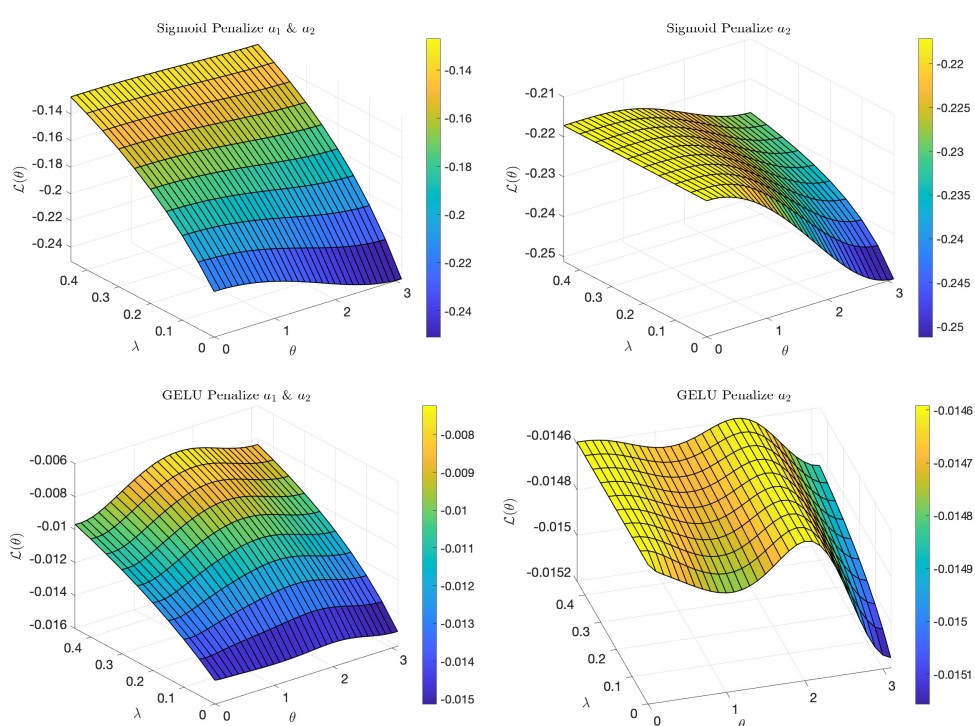

Figure 4: Graphs of the landscapes with $d = 2$ for sigmoid and GELU activation functions.

and boundary conditions, including higher-order or nonlinear equations, where feature structures may be more intricate. Beyond the deep Ritz method, it would also be valuable to explore whether optimizing losses defined by physics-informed neural networks exhibit analogous mechanisms of feature emergence. We plan to study questions along these directions in future work.

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
