APPENDIX

# A    PROOF OF THEOREM 2.2

In this section, we provide the proof of Theorem 2.2. First recall that

$$\mathcal{L}(\boldsymbol{w}) :=\mathbb{E}_{\boldsymbol{x}\sim\mathcal{P}_\Omega}\left[|\nabla_{\boldsymbol{x}}u_{\boldsymbol{w}}(\boldsymbol{x}) - \nabla_{\boldsymbol{x}}u^*(\boldsymbol{x})|^2 + |u_{\boldsymbol{w}}(\boldsymbol{x}) - u^*(\boldsymbol{x})|^2\right]$$

$$=\mathbb{E}_{\boldsymbol{x}\sim\mathcal{P}_\Omega}\left[|2\sigma(\boldsymbol{w}\cdot\boldsymbol{x})\boldsymbol{w} - \nabla_{\boldsymbol{x}}u^*(\boldsymbol{x})|^2 + |\sigma^2(\boldsymbol{w}\cdot\boldsymbol{x}) - u^*(\boldsymbol{x})|^2\right],$$

$$g_{\mathcal{L}}(\boldsymbol{w}) :=\mathrm{P}_{\boldsymbol{w}^\perp}\nabla\mathcal{L}(\boldsymbol{w})$$

$$=4\mathbb{E}_{\boldsymbol{x}\sim\mathcal{P}_\Omega}\left[\sigma'(\boldsymbol{w}\cdot\boldsymbol{x})\boldsymbol{w}^\top(2\sigma(\boldsymbol{w}\cdot\boldsymbol{x})\boldsymbol{w} - \nabla u^*)\mathrm{P}_{\boldsymbol{w}^\perp}\boldsymbol{x} - \sigma(\boldsymbol{w}\cdot\boldsymbol{x})\mathrm{P}_{\boldsymbol{w}^\perp}\nabla u^* + (\sigma^2(\boldsymbol{w}\cdot\boldsymbol{x}) - u^*(\boldsymbol{x}))\sigma(\boldsymbol{w}\cdot\boldsymbol{x})\mathrm{P}_{\boldsymbol{w}^\perp}\boldsymbol{x}\right],$$

and

$$\mathrm{OPT} := \mathcal{L}(\boldsymbol{w}^*) = \mathbb{E}_{\boldsymbol{x}\sim\mathcal{P}_\Omega}\left[|\nabla_{\boldsymbol{x}}u_{\boldsymbol{w}*}(\boldsymbol{x}) - \nabla_{\boldsymbol{x}}u^*(\boldsymbol{x})|^2 + |u_{\boldsymbol{w}*}(\boldsymbol{x}) - u^*(\boldsymbol{x})|^2\right],$$

where $\boldsymbol{w}^* := \arg\min_{\boldsymbol{w}\in\mathcal{S}^{d-1}}\mathcal{L}(\boldsymbol{w})$. With the above definitions and by Young's inequality, we can get

$$\mathcal{L}(\boldsymbol{w}) \leqslant 2\mathcal{L}^*(\boldsymbol{w}) + 2\mathrm{OPT},$$

where $\mathcal{L}^*(\boldsymbol{w})$ represents the loss in realizable setting and is defined as

$$\mathcal{L}^*(\boldsymbol{w}) :=\mathbb{E}_{\boldsymbol{x}\sim\mathcal{P}_\Omega}\left[|\nabla_{\boldsymbol{x}}u_{\boldsymbol{w}}(\boldsymbol{x}) - \nabla_{\boldsymbol{x}}u_{\boldsymbol{w}*}(\boldsymbol{x})|^2 + |u_{\boldsymbol{w}}(\boldsymbol{x}) - u_{\boldsymbol{w}*}(\boldsymbol{x})|^2\right]$$

$$=\mathbb{E}_{\boldsymbol{x}\sim\mathcal{P}_\Omega}\left[|2\sigma(\boldsymbol{w}\cdot\boldsymbol{x})\boldsymbol{w} - 2\sigma(\boldsymbol{w}^*\cdot\boldsymbol{x})\boldsymbol{w}^*|^2 + |\sigma^2(\boldsymbol{w}\cdot\boldsymbol{x}) - \sigma^2(\boldsymbol{w}^*\cdot\boldsymbol{x})|^2\right]. \tag{13}$$

The Riemannian gradient of the loss (13) is defined as

$$g^*(\boldsymbol{w}) :=\mathrm{P}_{\boldsymbol{w}^\perp}\nabla\mathcal{L}^*(\boldsymbol{w})$$

$$=\mathbb{E}_{\boldsymbol{x}\sim\mathcal{P}_\Omega}4\left[\sigma'(\boldsymbol{w}\cdot\boldsymbol{x})\boldsymbol{w}^\top(2\sigma(\boldsymbol{w}\cdot\boldsymbol{x})\boldsymbol{w} - 2\sigma(\boldsymbol{w}^*\cdot\boldsymbol{x})\boldsymbol{w}^*)\mathrm{P}_{\boldsymbol{w}^\perp}\boldsymbol{x} - 2\sigma(\boldsymbol{w}\cdot\boldsymbol{x})\sigma(\boldsymbol{w}^*\cdot\boldsymbol{x})\mathrm{P}_{\boldsymbol{w}^\perp}\boldsymbol{w}^*\right]$$

$$+ \mathbb{E}_{\boldsymbol{x}\sim\mathcal{P}_\Omega}\left[4(\sigma^2(\boldsymbol{w}\cdot\boldsymbol{x}) - \sigma^2(\boldsymbol{w}^*\cdot\boldsymbol{x}))\sigma(\boldsymbol{w}\cdot\boldsymbol{x})\mathrm{P}_{\boldsymbol{w}^\perp}\boldsymbol{x}\right].$$

We denote the difference between the Riemannian gradient of the agnostic loss $g_{\mathcal{L}}(\boldsymbol{w})$ and the realizable loss $g^*(\boldsymbol{w})$ by $\xi(\boldsymbol{w})$, i.e.,

$$\xi(\boldsymbol{w}) :=g_{\mathcal{L}}(\boldsymbol{w}) - g^*(\boldsymbol{w})$$

$$=4\mathbb{E}_{\boldsymbol{x}\sim\mathcal{P}_\Omega}\left[\sigma'(\boldsymbol{w}\cdot\boldsymbol{x})\boldsymbol{w}^\top(2\sigma(\boldsymbol{w}^*\cdot\boldsymbol{x})\boldsymbol{w}^* - \nabla u^*)\mathrm{P}_{\boldsymbol{w}^\perp}\boldsymbol{x} + \sigma(\boldsymbol{w}\cdot\boldsymbol{x})(2\sigma(\boldsymbol{w}^*\cdot\boldsymbol{x})P_{\boldsymbol{w}^\perp}\boldsymbol{w}^* - \mathrm{P}_{\boldsymbol{w}^\perp}\nabla u^*)\right]$$

$$+ 4\mathbb{E}_{\boldsymbol{x}\sim\mathcal{P}_\Omega}\left[(\sigma^2(\boldsymbol{w}^*\cdot\boldsymbol{x}) - u^*)\sigma(\boldsymbol{w}\cdot\boldsymbol{x})\mathrm{P}_{\boldsymbol{w}^\perp}\boldsymbol{x}\right]. \tag{14}$$

Throughout this section, we define a constant $C_d := \frac{8}{\sqrt{d+2}}$. We first show that the norm of $\xi(\boldsymbol{w})$ and the inner product of $\xi(\boldsymbol{w})$ and $\boldsymbol{w}^*$ are bounded.

**Lemma A.1.** *Let $\xi(\boldsymbol{w})$ as defined in (14). Then,*

$$|\xi(\boldsymbol{w})|_2 \leqslant C_d\sqrt{OPT} \quad \text{and} \quad |\xi(\boldsymbol{w})\cdot\boldsymbol{w}^*| \leqslant C_d\sqrt{OPT}|(\boldsymbol{w}^*)^{\perp\boldsymbol{w}}|_2.$$

*Proof.* For simplicity, we use $\mathbb{E}$ to denote $\mathbb{E}_{\boldsymbol{x}\sim\mathcal{P}_\Omega}$ throughout this proof. By the definition of $\xi(\boldsymbol{w})$ and the definition of the 2-norm, we have

$$|\xi(\boldsymbol{w})|_2$$

$$= \max_{\boldsymbol{v}\in\mathcal{S}^{d-1}}\mathbb{E}\left[4\sigma'(\boldsymbol{w}\cdot\boldsymbol{x})\boldsymbol{w}^\top(2\sigma(\boldsymbol{w}^*\cdot\boldsymbol{x})\boldsymbol{w}^* - \nabla_{\boldsymbol{x}}u^*)\mathrm{P}_{\boldsymbol{w}^\perp}\boldsymbol{x}\cdot\boldsymbol{v} + 4\sigma(\boldsymbol{w}\cdot\boldsymbol{x})(2\sigma(\boldsymbol{w}^*\cdot\boldsymbol{x})\mathrm{P}_{\boldsymbol{w}^\perp}\boldsymbol{w}^* - \mathrm{P}_{\boldsymbol{w}^\perp}\nabla_{\boldsymbol{x}}u^*)\cdot\boldsymbol{v}\right]$$

$$+ \mathbb{E}\left[4(\sigma^2(\boldsymbol{w}^*\cdot\boldsymbol{x}) - u^*)\sigma(\boldsymbol{w}\cdot\boldsymbol{x})\sigma'(\boldsymbol{w}^\top\boldsymbol{x})\mathrm{P}_{\boldsymbol{w}^\perp}\boldsymbol{x}\cdot\boldsymbol{v}\right]$$

$$= \max_{\boldsymbol{v}\in\mathcal{S}^{d-1}}\mathbb{E}\left[4\sigma'(\boldsymbol{w}\cdot\boldsymbol{x})\boldsymbol{w}^\top(2\sigma(\boldsymbol{w}^*\cdot\boldsymbol{x})\boldsymbol{w}^* - \nabla_{\boldsymbol{x}}u^*)\boldsymbol{x}\cdot\mathrm{P}_{\boldsymbol{w}^\perp}\boldsymbol{v} + 4\sigma(\boldsymbol{w}\cdot\boldsymbol{x})(2\sigma(\boldsymbol{w}^*\cdot\boldsymbol{x})\mathrm{P}_{\boldsymbol{w}^\perp}\boldsymbol{w}^* - \mathrm{P}_{\boldsymbol{w}^\perp}\nabla_{\boldsymbol{x}}u^*)\cdot\boldsymbol{v}\right]$$

$$+ \mathbb{E}\left[4(\sigma^2(\boldsymbol{w}^*\cdot\boldsymbol{x}) - u^*)\sigma(\boldsymbol{w}\cdot\boldsymbol{x})\sigma'(\boldsymbol{w}^\top\boldsymbol{x})\boldsymbol{x}\cdot\mathrm{P}_{\boldsymbol{w}^\perp}\boldsymbol{v}\right]$$

$$\leqslant \max_{\boldsymbol{v}\in\mathcal{S}^{d-1}}\sqrt{\mathbb{E}\left[(\boldsymbol{w}^\top(2\sigma(\boldsymbol{w}^*\cdot\boldsymbol{x})\boldsymbol{w}^* - \nabla_{\boldsymbol{x}}u^*))^2\right]\mathbb{E}\left[(4\sigma'(\boldsymbol{w}\cdot\boldsymbol{x})\boldsymbol{x}\cdot\mathrm{P}_{\boldsymbol{w}^\perp}\boldsymbol{v})^2\right]}$$

$$+ \sqrt{\mathbb{E}\left[((2\sigma(\boldsymbol{w}^*\cdot\boldsymbol{x})\boldsymbol{w}^* - \nabla_{\boldsymbol{x}}u^*)\cdot\mathrm{P}_{\boldsymbol{w}^\perp}\boldsymbol{v})^2\right]\mathbb{E}\left[(4\sigma(\boldsymbol{w}\cdot\boldsymbol{x}))^2\right]}$$

$$+ \sqrt{\mathbb{E}\left[(\sigma^2(\boldsymbol{w}^*\cdot\boldsymbol{x}) - u^*)^2\right]\mathbb{E}\left[(4\sigma(\boldsymbol{w}\cdot\boldsymbol{x})\sigma'(\boldsymbol{w}^\top\boldsymbol{x})\boldsymbol{x}\cdot\mathrm{P}_{\boldsymbol{w}^\perp}\boldsymbol{v})^2\right]}$$

$$\leqslant \left(\frac{4}{\sqrt{2(d+2)}} + \frac{4}{\sqrt{2(d+2)}}\right)\sqrt{\mathbb{E}\left[|2\sigma(\boldsymbol{w}^*\cdot\boldsymbol{x})\boldsymbol{w}^* - \nabla_{\boldsymbol{x}}u^*|^2\right]} + \frac{4}{\sqrt{2(d+2)(d+4)}}\sqrt{\mathbb{E}\left[|\sigma^2(\boldsymbol{w}^*\cdot\boldsymbol{x}) - u^*|^2\right]}.$$

Recall that

$$\text{OPT} := \mathbb{E}_{\boldsymbol{x}\sim\mathcal{P}_\Omega}\left[|\nabla_{\boldsymbol{x}}u_{\boldsymbol{w}^*}(\boldsymbol{x}) - \nabla_{\boldsymbol{x}}u^*(\boldsymbol{x})|^2 + |u_{\boldsymbol{w}^*}(\boldsymbol{x}) - u^*(\boldsymbol{x})|^2\right]$$
$$= \mathbb{E}_{\boldsymbol{x}\sim\mathcal{P}_\Omega}\left[|2\sigma(\boldsymbol{w}^*\cdot\boldsymbol{x})\boldsymbol{w}^* - \nabla_{\boldsymbol{x}}u^*|^2 + |\sigma^2(\boldsymbol{w}^*\cdot\boldsymbol{x}) - u^*|^2\right].$$

Hence,

$$|\xi(\boldsymbol{w})|_2 \leqslant \frac{8}{\sqrt{2(d+2)}}\sqrt{2\text{OPT}} = C_d\sqrt{\text{OPT}}.$$

Similarly, we can get

$$|\xi(\boldsymbol{w})\cdot\boldsymbol{w}^*| \leqslant C_d\sqrt{\text{OPT}}|(\boldsymbol{w}^*)^{\perp\boldsymbol{w}}|_2.$$

$\square$

Then, by the triangle inequality, we have the following Corollary.

**Corollary A.2.** *For any* $\boldsymbol{w}\in\mathcal{S}^{d-1}$, $|g_\mathcal{L}(\boldsymbol{w})| \leqslant |\xi(\boldsymbol{w})| + |g^*(\boldsymbol{w})| \leqslant C_d\sqrt{OPT} + |g^*(\boldsymbol{w})|$.

With the above results, we are ready to show the sharpness property of the Riemannian gradient $g_\mathcal{L}(\boldsymbol{w})$. Before that, we first note that the expectation in $\mathcal{L}^*(\boldsymbol{w})$ can be explicitly calculated by applying the similar techniques as in Cho & Saul (2009). We calculate the expectations as

$$\mathbb{E}_{\boldsymbol{x}\sim\mathcal{P}_\Omega}[\sigma(\boldsymbol{w}\cdot\boldsymbol{x})\sigma(\boldsymbol{w}^*\cdot\boldsymbol{x})\boldsymbol{w}\cdot\boldsymbol{w}^*] = \frac{1}{2\pi(d+2)}\left(\sin\theta\cos\theta + (\pi-\theta)\cos^2\theta\right),$$

$$\mathbb{E}_{\boldsymbol{x}\sim\mathcal{P}_\Omega}[\sigma^2(\boldsymbol{w}\cdot\boldsymbol{x})\sigma^2(\boldsymbol{w}^*\cdot\boldsymbol{x})] = \frac{1}{2\pi(d+2)(d+4)}(3\sin\theta\cos\theta + (\pi-\theta)(1 + 2\cos^2\theta)),$$

where $\theta$ is the angle between $\boldsymbol{w}$ and $\boldsymbol{w}^*$ and $\theta\in[0,\pi]$. Then we have

$$\mathcal{L}^*(\boldsymbol{w}) = \left(\frac{4}{d+2} - \frac{4}{\pi(d+2)}\left(\sin\theta\cos\theta + (\pi-\theta)\cos^2\theta\right)\right)$$
$$+ \left(\frac{3}{(d+2)(d+4)} - \frac{1}{\pi(d+2)(d+4)}\left(3\sin\theta\cos\theta + (\pi-\theta)(1 + 2\cos^2\theta)\right)\right).$$

The sharpness property of $g_\mathcal{L}(\boldsymbol{w})$ is shown as follows.

**Lemma A.3** (Sharpness). *Let* $\theta := \angle(\boldsymbol{w},\boldsymbol{w}^*)$ *(angle between* $\boldsymbol{w}$ *and* $\boldsymbol{w}^*$*). Assume* $\theta\in[0,\frac{\pi}{2}]$ *and* $\sin\theta \geqslant \frac{32\pi\sqrt{OPT}}{C_d}$*, then*

$$g_\mathcal{L}(\boldsymbol{w})\cdot\boldsymbol{w}^* \leqslant -\frac{1}{2}|g^*(\boldsymbol{w})|_2\sin\theta.$$

*Proof.* We take the Riemannian gradient of $\mathcal{L}^*(\boldsymbol{w})$ to get

$$g^*(\boldsymbol{w}) = \left(-\frac{4(\sin\theta + 2(\pi-\theta)\cos\theta)}{\pi(d+2)} - \frac{4\sin\theta + 4(\pi-\theta)\cos\theta}{\pi(d+2)(d+4)}\right)\mathrm{P}_{\boldsymbol{w}^\perp}\boldsymbol{w}^*.$$

By noting that

$$g^*(\boldsymbol{w})\cdot\boldsymbol{w}^* = \left(-\frac{4(\sin\theta + 2(\pi-\theta)\cos\theta)}{\pi(d+2)} - \frac{4\sin\theta + 4(\pi-\theta)\cos\theta}{\pi(d+2)(d+4)}\right)\mathrm{P}_{\boldsymbol{w}^\perp}\boldsymbol{w}^*\cdot\boldsymbol{w}^*$$
$$= -|g^*(\boldsymbol{w})|_2\sin\theta,$$

and applying Lemma A.1, we can get

$$g_\mathcal{L}(\boldsymbol{w})\cdot\boldsymbol{w}^* = g^*(\boldsymbol{w})\cdot\boldsymbol{w}^* + \xi(\boldsymbol{w})\cdot\boldsymbol{w}^* \leqslant -\left(|g^*(\boldsymbol{w})|_2 - C_d\sqrt{\text{OPT}}\right)\sin\theta.$$

Then we consider $|g^*(\boldsymbol{w})|$. We have that

$$|g^*(\boldsymbol{w})|_2 = \left(\frac{4(\sin\theta + 2(\pi-\theta)\cos\theta)}{\pi(d+2)} + \frac{4\sin\theta + 4(\pi-\theta)\cos\theta}{\pi(d+2)(d+4)}\right)\sin\theta,$$

and note that when $\theta\in[0,\frac{\pi}{2}]$,

$$\underline{c}\sin\theta \leqslant |g^*(\boldsymbol{w})| \leqslant \bar{c}\sin\theta,$$

where $\bar{c} := \frac{8}{(d+2)} + \frac{4}{(d+2)(d+4)}$ and $\underline{c} := \frac{4}{\pi(d+2)} + \frac{4}{\pi(d+2)(d+4)}$. Therefore, by noting that $\underline{c} \geqslant \frac{C_d^2}{16\pi}$, we have that when $\sin\theta \geqslant \frac{2C_d\sqrt{\text{OPT}}}{C_d^2/(16\pi)} \geqslant \frac{2C_d\sqrt{\text{OPT}}}{\underline{c}}$,

$$|g^*(\boldsymbol{w})|_2 \geqslant 2C_d\sqrt{\text{OPT}}, \quad \frac{1}{2}|g^*(\boldsymbol{w})|_2 \geqslant C_d\sqrt{\text{OPT}}.$$

Hence, we can get

$$|g^*(\boldsymbol{w})|_2 - C_d\sqrt{\text{OPT}} \geqslant |g^*(\boldsymbol{w})|_2 - \frac{1}{2}|g^*(\boldsymbol{w})|_2 = \frac{1}{2}|g^*(\boldsymbol{w})|_2,$$

and

$$g_{\mathcal{L}}(\boldsymbol{w}) \cdot \boldsymbol{w}^* \leqslant -\frac{1}{2}|g^*(\boldsymbol{w})|_2 \sin\theta.$$

$\square$

If $\sin\theta < \frac{32\pi\sqrt{\text{OPT}}}{C_d}$, we can get the approximate optimal solution immediately, which is shown in the following lemma.

**Lemma A.4.** *Let $\theta := \angle(\boldsymbol{w}, \boldsymbol{w}^*)$. If $\theta \in [0, \frac{\pi}{2}]$ and $\sin\theta < \frac{32\pi\sqrt{\text{OPT}}}{C_d}$, then*

$$\mathcal{L}(\boldsymbol{w}) < (256\pi^2 + 2)OPT.$$

*Proof.* By Young's inequality, we have

$$\begin{aligned}
\mathcal{L}(\boldsymbol{w}) \leqslant{} & 2\text{OPT} + 2\mathcal{L}^*(\boldsymbol{w}) \\
={} & 2\text{OPT} + \frac{8}{d+2}\left(1 - \frac{1}{\pi}\sin\theta\cos\theta - (1 - \frac{\theta}{\pi})\cos^2\theta\right) \\
& + \frac{2}{(d+2)(d+4)}\left(3 - \frac{3}{\pi}\sin\theta\cos\theta - (1 - \frac{\theta}{\pi})(1 + 2\cos^2\theta)\right) \\
\leqslant{} & 2\text{OPT} + \frac{8}{d+2}\left(\sin^2\theta + \frac{1}{\pi}\cos\theta(\theta\cos\theta - \sin\theta)\right) \\
& + \frac{2}{(d+2)(d+4)}\left(3\sin^2\theta + \frac{3}{\pi}\cos\theta(\theta\cos\theta - \sin\theta)\right) \\
\leqslant{} & 2\text{OPT} + \frac{8}{d+2}\sin^2\theta + \frac{6}{(d+2)(d+4)}\sin^2\theta \\
\leqslant{} & \left(\left(\frac{8}{d+2} + \frac{6}{(d+2)(d+4)}\right)\frac{(32\pi)^2}{C_d^2} + 2\right)\text{OPT},
\end{aligned}$$

where the second inequality comes from $\frac{\theta}{\pi} - 1 \leqslant 0$ and the third inequality comes from $\theta\cos\theta - \sin\theta \leqslant 0$. Since $\frac{8}{d+2} + \frac{6}{(d+2)(d+4)} < \frac{C_d^2}{4}$, we have

$$\mathcal{L}(\boldsymbol{w}) < (256\pi^2 + 2)\text{OPT}.$$

$\square$

Next, we provide the uniform upper bound on the number of samples required to approximate the Riemannian gradients $g_{\mathcal{E}}(\boldsymbol{w})$ in Lemma A.5 with the following definitions the empirical estimate of $g_{\mathcal{E}}(\boldsymbol{w})$:

$$\hat{g}_{\mathcal{E}}(\boldsymbol{w}) := \frac{1}{n}\sum_{i=1}^{n} g_{\mathcal{E}}(\boldsymbol{w}; \boldsymbol{x}_i),$$

$$g_{\mathcal{E}}(\boldsymbol{w}; \boldsymbol{x}_i) := 4\left(2|\boldsymbol{w}|^2\sigma(\boldsymbol{w}\cdot\boldsymbol{x}_i)\text{P}_{\boldsymbol{w}^\perp}\boldsymbol{x}_i + \sigma^3(\boldsymbol{w}\cdot\boldsymbol{x}_i)\text{P}_{\boldsymbol{w}^\perp}\boldsymbol{x}_i - f(\boldsymbol{x}_i)\sigma(\boldsymbol{w}\cdot\boldsymbol{x}_i)\text{P}_{\boldsymbol{w}^\perp}\boldsymbol{x}_i\right).$$

**Lemma A.5.** *Let $\boldsymbol{w}^*, \boldsymbol{w} \in \mathcal{S}^{d-1}$ and $\hat{g}_{\mathcal{E}}(\boldsymbol{w})$ be the empirical estimate of the Riemannian gradient $g_{\mathcal{E}}(\boldsymbol{w})$. Then, under Assumption 2.1, there exists a constant $C_1$ depending on $\delta, C_f$ such that with probability at least $1 - \delta$,*

$$\sup_{\boldsymbol{w} \in \mathcal{S}^{d-1}} |\hat{g}_{\mathcal{E}}(\boldsymbol{w}) - g_{\mathcal{E}}(\boldsymbol{w})| \lesssim \sqrt{\frac{C_1 d}{n}},$$

$$\sup_{\boldsymbol{w} \in \mathcal{S}^{d-1}} (\hat{g}_{\mathcal{E}}(\boldsymbol{w}) - g_{\mathcal{E}}(\boldsymbol{w})) \cdot \boldsymbol{w}^* \lesssim \sqrt{\frac{C_1 d}{n}}.$$

*Proof.* Let $\mathcal{G} := \{g_{\mathcal{E}}(\boldsymbol{w}) : \boldsymbol{w} \in \mathcal{S}^{d-1}\}$, then by Assumption 2.1 and that $|\boldsymbol{x}| \leqslant 1$, we have that $|g_{\mathcal{E}}(\boldsymbol{w}, \boldsymbol{x})| \leqslant M$ with $M := 12 + 4C_f$. Thanks to Theorem 4.10 of Wainwright (2019), one has that with probability at least $1 - \delta$,

$$\sup_{\boldsymbol{w} \in \mathcal{S}^{d-1}} |\hat{g}_{\mathcal{E}}(\boldsymbol{w}) - g_{\mathcal{E}}(\boldsymbol{w})| \leqslant 2\mathcal{R}_n(\mathcal{G}) + M\sqrt{\frac{2\log(1/\delta)}{n}}, \tag{15}$$

where $\mathcal{R}_n(\mathcal{G})$ is the Rademacher complexity of a function class $\mathcal{G}$, defined by

$$\mathcal{R}_n(\mathcal{G}) := \mathbb{E}_{\boldsymbol{x}_i, \varepsilon_i} \sup_{g \in \mathcal{G}} \left| \frac{1}{n} \sum_{i=1}^{n} \varepsilon_i g(\boldsymbol{x}_i) \right|,$$

where $\{\varepsilon_i\}_{i=1}^n$ is a sequence of i.i.d. Rademacher random variables. Moreover, under Assumption 2.1, one has that

$$\|g_{\mathcal{E}}(\boldsymbol{w}; \boldsymbol{x}) - g_{\mathcal{E}}(\boldsymbol{w}'; \boldsymbol{x})\|_\infty \leqslant L|\boldsymbol{w} - \boldsymbol{w}'|,$$

where $L := 64 + 16C_f$. This implies that

$$\mathcal{N}(\delta, \mathcal{G}, \|\cdot\|_\infty) \leqslant \mathcal{N}(\frac{\delta}{L}, \mathcal{S}^{d-1}, |\cdot|) \leqslant \left(\frac{3L}{\delta}\right)^d,$$

where $\mathcal{N}(\delta, \mathcal{G}, \|\cdot\|_\infty)$ denotes the $\delta$-covering number of $\mathcal{G}$ w.r.t. the $L_\infty$-norm and $\mathcal{N}(\frac{\delta}{L}, \mathcal{S}^{d-1}, |\cdot|)$ denotes the $\frac{\delta}{L}$-covering number of $\mathcal{S}^{d-1}$ w.r.t. the 2-norm. Applying Dudley's theorem (Wolf; Lu et al., 2021b) and noting that $\sup_{\boldsymbol{w} \in \mathcal{S}^{d-1}} \|g_{\mathcal{E}}(\boldsymbol{w})\|_\infty \leqslant M$, we can get

$$\mathcal{R}_n(\mathcal{G}) \leqslant \inf_{0 \leqslant \delta \leqslant M} \left\{ 4\delta + \frac{12}{\sqrt{n}} \int_\delta^M \sqrt{\log \mathcal{N}(\tau, \mathcal{G}, \|\cdot\|_\infty)} \, d\tau \right\}$$

$$\inf_{0 \leqslant \delta \leqslant M} \left\{ \leqslant 4\delta + \frac{12}{\sqrt{n}} \int_\delta^M \sqrt{d \log\left(\frac{3L}{\tau}\right)} \, d\tau \right\}.$$

By setting $\delta = 0$, we have

$$\mathcal{R}_n(\mathcal{G}) \leqslant 12\sqrt{\frac{d}{n}} \int_0^M \sqrt{\log(3L) + \log(1/\tau)} \, d\tau$$

$$\lesssim 12\sqrt{\frac{d}{n}} \int_0^M \sqrt{\log(3L)} + \sqrt{\log(1/\tau)} \, d\tau$$

$$\leqslant 12\sqrt{\frac{d}{n}} \left( M\sqrt{\log(3L)} + C \right)$$

$$\leqslant \sqrt{\frac{\tilde{C}_1 d}{n}},$$

where $\tilde{C}_1$ depends at mostly polynomially on $C_f$. Plugging the bound above into (15), we can conclude that with probability at least $1 - \delta$,

$$\sup_{\boldsymbol{w} \in \mathcal{S}^{d-1}} |\hat{g}_{\mathcal{E}}(\boldsymbol{w}) - g_{\mathcal{E}}(\boldsymbol{w})| \lesssim \sqrt{\frac{C_1 d}{n}}$$

and

$$\sup_{\boldsymbol{w} \in \mathcal{S}^{d-1}} (\hat{g}_{\mathcal{E}}(\boldsymbol{w}) - g_{\mathcal{E}}(\boldsymbol{w})) \cdot \boldsymbol{w}^* \leqslant \sup_{\boldsymbol{w} \in \mathcal{S}^{d-1}} |\hat{g}_{\mathcal{E}}(\boldsymbol{w}) - g_{\mathcal{E}}(\boldsymbol{w})|$$

$$\lesssim \sqrt{\frac{C_1 d}{n}}.$$

The constant $C_1$ depends on $C_f$ and $\delta$.

$\square$

With above lemmas and corollary, we proceed to show the main result of this subsection.

**Theorem A.6.** *Suppose that Assumption 2.1 holds. Consider Algorithm 1 with initial condition $\boldsymbol{w}^0$ satisfying $\angle(\boldsymbol{w}^0, \boldsymbol{w}^*) \in [0, \frac{\pi}{2}]$. If we choose the sample size $n = \Theta\left(\frac{C_1 d}{C_d^2 \epsilon}\right)$ and the step size $\eta = \frac{1}{32\pi C_d^2}$, then after $T = O(\log(1/\epsilon))$ iterations, with probability at least $1 - \delta$, the output of Algorithm 1 $\boldsymbol{w}^T$ satisfies $\mathcal{L}(\boldsymbol{w}^T) = O(OPT) + C_d^2 \epsilon$.*

*Proof.* Since $|\boldsymbol{w}^t - \eta \hat{g}_{\mathcal{E}}(\boldsymbol{w}^t)|^2 = 1 + \eta^2 |\hat{g}_{\mathcal{E}}(\boldsymbol{w}^t)|^2 \geqslant 1$, we have

$$
\begin{aligned}
|\boldsymbol{w}^{t+1} - \boldsymbol{w}^*|^2 &\leqslant |\boldsymbol{w}^t - \eta \hat{g}_{\mathcal{E}}(\boldsymbol{w}^t) - \boldsymbol{w}^*|^2 \\
&= |\boldsymbol{w}^t - \boldsymbol{w}^*|^2 + 2\eta \hat{g}_{\mathcal{E}}(\boldsymbol{w}^t) \cdot (\boldsymbol{w}^* - \boldsymbol{w}^t) + \eta^2 |\hat{g}_{\mathcal{E}}(\boldsymbol{w}^t)|^2.
\end{aligned}
\tag{16}
$$

Let us denote the angle between $\boldsymbol{w}^t$ and $\boldsymbol{w}^*$ by $\theta_t$ and assume that $\theta_t$ satisfies $\sin \theta_t \geqslant \frac{32\pi\sqrt{\text{OPT}}}{C_d} + \sqrt{\epsilon}$, hence the condition for Lemma A.3 is satisfied. Note by definition $\hat{g}_{\mathcal{E}}(\boldsymbol{w}^t) \perp \boldsymbol{w}^t$, hence using Lemma A.3 and Lemma A.5, we have that with probability at least $1 - \delta$,

$$
\begin{aligned}
\hat{g}_{\mathcal{E}}(\boldsymbol{w}^t) \cdot (\boldsymbol{w}^* - \boldsymbol{w}^t) &= \hat{g}_{\mathcal{E}}(\boldsymbol{w}^t) \cdot \boldsymbol{w}^* \\
&= (\hat{g}_{\mathcal{E}}(\boldsymbol{w}^t) - g_{\mathcal{L}}(\boldsymbol{w}^t)) \cdot \boldsymbol{w}^* + g_{\mathcal{L}}(\boldsymbol{w}^t) \cdot \boldsymbol{w}^* \\
&= (\hat{g}_{\mathcal{E}}(\boldsymbol{w}^t) - g_{\mathcal{E}}(\boldsymbol{w}^t)) \cdot \boldsymbol{w}^* + (g_{\mathcal{E}}(\boldsymbol{w}^t) - g_{\mathcal{L}}(\boldsymbol{w}^t)) \cdot \boldsymbol{w}^* + g_{\mathcal{L}}(\boldsymbol{w}^t) \cdot \boldsymbol{w}^* \\
&\leqslant \sqrt{\frac{C_1 d}{n}} - \frac{1}{2} |g^*(\boldsymbol{w}^t)|_2 \sin \theta_t,
\end{aligned}
\tag{17}
$$

by noting that $g_{\mathcal{E}}(\boldsymbol{w}^t) = g_{\mathcal{L}}(\boldsymbol{w}^t)$. By Corollary A.2 and Lemma A.5, the squared norm term $|\hat{g}(\boldsymbol{w}^t)|^2$ in equation (16) can be bounded by

$$
\begin{aligned}
|\hat{g}_{\mathcal{E}}(\boldsymbol{w}^t)|^2 &= |\hat{g}_{\mathcal{E}}(\boldsymbol{w}^t) - g_{\mathcal{L}}(\boldsymbol{w}^t) + g_{\mathcal{L}}(\boldsymbol{w}^t)|^2 \\
&\leqslant 2|\hat{g}_{\mathcal{E}}(\boldsymbol{w}^t) - g_{\mathcal{L}}(\boldsymbol{w}^t)|^2 + 2|g_{\mathcal{L}}(\boldsymbol{w}^t)|^2 \\
&\leqslant 4|\hat{g}_{\mathcal{E}}(\boldsymbol{w}^t) - g_{\mathcal{E}}(\boldsymbol{w}^t)|^2 + 4|g_{\mathcal{E}}(\boldsymbol{w}^t) - g_{\mathcal{L}}(\boldsymbol{w}^t)|^2 + 2|g_{\mathcal{L}}(\boldsymbol{w}^t)|^2 \\
&\leqslant \frac{4C_1 d}{n} + 4C_d^2 \text{OPT} + 4|g^*(\boldsymbol{w}^t)|^2,
\end{aligned}
\tag{18}
$$

by noting that $g_{\mathcal{E}}(\boldsymbol{w}^t) = g_{\mathcal{L}}(\boldsymbol{w}^t)$. Plugging (17) and (18) back into (16) and letting $\kappa_d := \sqrt{4C_1 d}$, we have that with probability at least $1 - \delta$,

$$
\begin{aligned}
|\boldsymbol{w}^{t+1} - \boldsymbol{w}^*|^2 &\leqslant |\boldsymbol{w}^t - \boldsymbol{w}^*|^2 + 2\eta \left(\frac{\kappa_d}{\sqrt{n}} - \frac{1}{2}|g^*(\boldsymbol{w}^t)| \sin \theta_t\right) + \eta^2 \left(\frac{\kappa_d^2}{n} + 4C_d^2 \text{OPT} + 4|g^*(\boldsymbol{w}^t)|^2\right) \\
&= |\boldsymbol{w}^t - \boldsymbol{w}^*|^2 + \frac{2\eta\kappa_d}{\sqrt{n}} - \eta|g^*(\boldsymbol{w}^t)| \sin \theta_t + \eta^2 \left(\frac{\kappa_d^2}{n} + 4C_d^2 \text{OPT} + 4|g^*(\boldsymbol{w}^t)|^2\right).
\end{aligned}
$$

We first assume that $\theta_t \leqslant \theta_{t-1} \leqslant \cdots \leqslant \theta_0 \leqslant \delta = \frac{\pi}{2}$ and $\sin \theta_t \geqslant \frac{32\pi\sqrt{\text{OPT}}}{C_d} + \sqrt{\epsilon}$. We will argue that $\theta_{t+1} \leqslant \theta_t$ in this case. Then, by an inductive argument, we immediately know that the assumption is valid and that $\theta_t$ is a decreasing sequence (as long as $\sin \theta_t \geqslant \frac{32\pi\sqrt{\text{OPT}}}{C_d} + \sqrt{\epsilon}$). To prove $\theta_{t+1} \leqslant \theta_t$, we first recall that

$$|g^*(\boldsymbol{w}^t)|_2 = \left(\frac{4(\sin \theta_t + 2(\pi - \theta_t)\cos \theta_t)}{\pi(d+2)} + \frac{4\sin \theta_t + 4(\pi - \theta_t)\cos \theta_t}{\pi(d+2)(d+4)}\right) \sin \theta_t,$$

and

$$\underline{c}\sin\theta_t \leqslant |g^*(\boldsymbol{w}^t)| \leqslant \bar{c}\sin\theta_t,$$

where $\bar{c} := \frac{8}{(d+2)} + \frac{4}{(d+2)(d+4)}$ and $\underline{c} := \frac{4}{\pi(d+2)} + \frac{4}{\pi(d+2)(d+4)}$. Noting that $C_d^2 < 8\bar{c}$, we have

$$|\boldsymbol{w}^{t+1} - \boldsymbol{w}^*|^2 \leqslant |\boldsymbol{w}^t - \boldsymbol{w}^*|^2 + \frac{2\eta\kappa_d}{\sqrt{n}} - \eta\underline{c}\sin^2\theta_t + \eta^2\left(\frac{\kappa_d^2}{n} + 4C_d^2\text{OPT} + 4\bar{c}^2\sin^2\theta_t\right)$$

$$< |\boldsymbol{w}^t - \boldsymbol{w}^*|^2 + \frac{2\eta\kappa_d}{\sqrt{n}} - \eta\underline{c}\sin^2\theta_t + 4\eta^2\bar{c}^2\left(\frac{\kappa_d^2}{4n\bar{c}^2} + \frac{8}{\bar{c}}\text{OPT} + \sin^2\theta_t\right).$$

Now choosing $n \gtrsim \kappa_d^2/(4\bar{c}^2\epsilon)$ and recalling that

$$\sin^2\theta_t \geqslant \left(\frac{32\pi\sqrt{\text{OPT}}}{C_d} + \sqrt{\epsilon}\right)^2 \geqslant \frac{(32\pi)^2\text{OPT}}{C_d^2} + \epsilon > \frac{8}{\bar{c}}\text{OPT} + \epsilon,$$

we can further bound $|\boldsymbol{w}^{t+1} - \boldsymbol{w}^*|_2^2$ above as

$$|\boldsymbol{w}^{t+1} - \boldsymbol{w}^*|^2 < |\boldsymbol{w}^t - \boldsymbol{w}^*|^2 + \frac{2\eta\kappa_d}{\sqrt{n}} - \eta\underline{c}\sin^2\theta_t + 8\eta^2\bar{c}^2\sin^2\theta_t \tag{19}$$

$$< |\boldsymbol{w}^t - \boldsymbol{w}^*|^2 - \eta\underline{c}\sin^2\theta_t + 8\eta^2\bar{c}^2\sin^2\theta_t.$$

Recalling that

$$|\boldsymbol{w}^t - \boldsymbol{w}^*|^2 = 2 - 2\cos\theta_t = 4\sin^2(\theta_t/2),$$

and for any $\theta_t \leqslant \delta = \frac{\pi}{2}$, we have $\sqrt{2}\sin(\theta_t/2) \leqslant \sin\theta_t \leqslant 2\sin(\theta_t/2)$, hence

$$|\boldsymbol{w}^{t+1} - \boldsymbol{w}^*|^2 < \left(1 - \frac{\eta\underline{c}}{2} + 8\eta^2\bar{c}^2\right)|\boldsymbol{w}^t - \boldsymbol{w}^*|^2.$$

Noting that $\bar{c} < \frac{C_d^2}{4}, \underline{c} > \frac{C_d^2}{16\pi}$ and choosing $\eta = \frac{1}{32\pi C_d^2}$ yields

$$4\sin^2(\theta_{t+1}/2) = |\boldsymbol{w}^{t+1} - \boldsymbol{w}^*|^2$$

$$< \left(1 - \frac{1}{2048\pi^2}\right)|\boldsymbol{w}^t - \boldsymbol{w}^*|^2 \tag{20}$$

$$= \left(1 - \frac{1}{2048\pi^2}\right)(4\sin^2(\theta_t/2)).$$

This shows that $\theta_{t+1} \leqslant \theta_t$, hence completing the inductive argument. Furthermore, (20) implies that after at most $T = O(\log(1/\epsilon))$ iterations, it must hold that $\sin\theta_T \leqslant \frac{32\pi\sqrt{\text{OPT}}}{C_d} + \sqrt{\epsilon}$. Although (20) only holds when $\sin\theta_T \geqslant \frac{32\pi\sqrt{\text{OPT}}}{C_d} + \sqrt{\epsilon}$, we can further show that if after some iterations $t^*$ we have $\sin\theta_{t^*} \leqslant \frac{32\pi\sqrt{\text{OPT}}}{C_d} + \sqrt{\epsilon}$, then $\sin\theta_{t^*+1}$ is still of order $\sqrt{\text{OPT}} + \sqrt{\epsilon}$. Concretely, if there exists some step $t^* \leqslant T$ such that $\sin\theta_{t^*} \leqslant \frac{32\pi\sqrt{\text{OPT}}}{C_d} + \sqrt{\epsilon}$, then at step $t^* + 1$ it must hold (by (19)):

$$\sin\theta_{t^*+1} \leqslant \sqrt{2 + 8\eta^2\bar{c}^2}\sin\theta_{t^*} \leqslant 2\sin\theta_{t^*} \leqslant 2\left(\frac{32\pi\sqrt{\text{OPT}}}{C_d} + \sqrt{\epsilon}\right)$$

In other words, for all steps $t^* \leqslant t \leqslant T$, it holds that $\sin \theta_t \leqslant 2 \left( \frac{32\pi\sqrt{\text{OPT}}}{C_d} + \sqrt{\epsilon} \right)$. Recall that by Young's inequality, we have

$$
\begin{aligned}
\mathcal{L}(\boldsymbol{w}^t) \leqslant & 2\text{OPT} + 2\mathcal{L}^*(\boldsymbol{w}^t) \\
= & 2\text{OPT} + \frac{8}{d+2} \left( 1 - \frac{1}{\pi} \sin \theta_t \cos \theta_t - (1 - \frac{\theta_t}{\pi}) \cos^2 \theta_t \right) \\
& + \frac{2}{(d+2)(d+4)} \left( 3 - \frac{3}{\pi} \sin \theta_t \cos \theta_t - (1 - \frac{\theta_t}{\pi})(1 + 2\cos^2 \theta_t) \right) \\
\leqslant & 2\text{OPT} + \frac{8}{d+2} \left( \sin^2 \theta_t + \frac{1}{\pi} \cos \theta_t (\theta_t \cos \theta_t - \sin \theta_t) \right) \\
& + \frac{2}{(d+2)(d+4)} \left( 3\sin^2 \theta_t + \frac{3}{\pi} \cos \theta_t (\theta_t \cos \theta_t - \sin \theta_t) \right) \\
\leqslant & 2\text{OPT} + \frac{8}{d+2} \sin^2 \theta_t + \frac{6}{(d+2)(d+4)} \sin^2 \theta_t \\
\leqslant & \left( 8 \left( \frac{8}{d+2} + \frac{6}{(d+2)(d+4)} \right) \frac{(32\pi)^2}{C_d^2} + 2 \right) \text{OPT} + 8 \left( \frac{8}{d+2} + \frac{6}{(d+2)(d+4)} \right) \epsilon.
\end{aligned}
$$

Since $\frac{8}{d+2} + \frac{6}{(d+2)(d+4)} < \frac{C_d^2}{4}$, we have

$$
\mathcal{L}(\boldsymbol{w}^t) < (2048\pi^2 + 2)\text{OPT} + 2C_d^2 \epsilon \lesssim \text{OPT} + 2C_d^2 \epsilon.
$$

Thus, in summary, choosing $T = O(\log(1/\epsilon))$, we get that with probability at least $1 - \delta$, $\sin \theta_T \lesssim \frac{\sqrt{\text{OPT}}}{C_d} + \sqrt{\epsilon}$. Also, Riemannian GD (Algorithm 1) outputs $\boldsymbol{w}^T$ such that with probability at least $1 - \delta$, $\mathcal{L}(\boldsymbol{w}^T) = O(\text{OPT}) + 2C_d^2 \epsilon$, with sample size $n = \Theta \left( \frac{C_1 d}{C_d^2 \epsilon} \right)$. $\qquad \square$

## B  PROOF OF PROPOSITION 2.3

In this section, we present the proof of Proposition 2.3.

*Proof.* Applying similar arguments as in Cho & Saul (2009), one can obtain that

$$
\mathbb{E}_{\boldsymbol{x} \sim \mathcal{P}_\Omega}[\sigma(\boldsymbol{w}^* \cdot \boldsymbol{x})\sigma(\boldsymbol{w} \cdot \boldsymbol{x})] = \frac{1}{2\pi(d+2)}(\sin \theta + (\pi - \theta) \cos \theta),
$$

$$
\mathbb{E}_{\boldsymbol{x} \sim \mathcal{P}_\Omega}[\sigma^2(\boldsymbol{w}^* \cdot \boldsymbol{x})\sigma^2(\boldsymbol{w} \cdot \boldsymbol{x})] = \frac{1}{2\pi(d+2)(d+4)}(3\sin \theta \cos \theta + (\pi - \theta)(1 + 2\cos^2 \theta)).
$$

where $\theta := \angle(\boldsymbol{w}, \boldsymbol{w}^*)$. Hence, we have

$$
\mathbb{E}_{\boldsymbol{x} \sim \mathcal{P}_\Omega}[|2\sigma(\boldsymbol{w} \cdot \boldsymbol{x})\boldsymbol{w}|^2] = \frac{2}{d+2}, \quad \mathbb{E}_{\boldsymbol{x} \sim \mathcal{P}_\Omega}[|\sigma^2(\boldsymbol{w} \cdot \boldsymbol{x})|^2] = \frac{3}{2(d+2)(d+4)}.
$$

Therefore, $\min_{\boldsymbol{w} \in \mathcal{S}^{d-1}} \mathcal{E}(\boldsymbol{w})$ is equivalent to $\max_{\boldsymbol{w} \in \mathcal{S}^{d-1}} \mathbb{E}_{\boldsymbol{x} \sim \mathcal{P}_\Omega}[\sigma^2(\boldsymbol{w}^* \cdot \boldsymbol{x})\sigma^2(\boldsymbol{w} \cdot \boldsymbol{x})]$, which is achieved when $\theta = 0$, i.e., $\boldsymbol{w} = \boldsymbol{w}^*$. $\qquad \square$

## C  PROOF OF LEMMA 2.4 AND THEOREM 2.5

In this section, we present the proof of Lemma 2.4 and Theorem 2.5. For completeness of the section, we include Lemma 2.4 and Theorem 2.5 again.

**Lemma C.1.** *For any fixed* $\boldsymbol{w}$*, the minimizer* $\boldsymbol{a}^*$ *of the regularized loss function (9) has a closed form* $\boldsymbol{a}^* = (\boldsymbol{K}_1 + \boldsymbol{K}_2 + \lambda \boldsymbol{I}_2)^{-1} \boldsymbol{K}_*$ *and* $\mathcal{L}_\lambda(\boldsymbol{w}) = -\boldsymbol{K}_*^\top (\boldsymbol{K}_1 + \boldsymbol{K}_2 + \lambda \boldsymbol{I}_2)^{-1} \boldsymbol{K}_*$.

*Proof.* Recall that

$$
\mathcal{L}_\lambda(\boldsymbol{w}, \boldsymbol{a}) = \boldsymbol{a}^\top \boldsymbol{K}_1 \boldsymbol{a} + \boldsymbol{a}^\top \boldsymbol{K}_2 \boldsymbol{a} - 2\boldsymbol{a}^\top \boldsymbol{K}_* + \lambda \boldsymbol{a}^\top \boldsymbol{a}.
$$

To find the minimizer $\boldsymbol{a}^*$, we take gradient of $\mathcal{L}_\lambda(\boldsymbol{w}, \boldsymbol{a})$ with respect to $\boldsymbol{a}$ and let the gradient be zero. The gradient of $\mathcal{L}_\lambda(\boldsymbol{w}, \boldsymbol{a})$ with respect to $\boldsymbol{a}$ is

$$\nabla_{\boldsymbol{a}} L_\lambda(\boldsymbol{w}, \boldsymbol{a}) = 2\boldsymbol{K}_1\boldsymbol{a} + 2\boldsymbol{K}_2\boldsymbol{a} - 2\boldsymbol{K}_* + 2\lambda\boldsymbol{I}_2\boldsymbol{a},$$

with $\boldsymbol{I}_2$ denoting the $2 \times 2$ identity matrix. By letting $\nabla_{\boldsymbol{a}} L_\lambda(\boldsymbol{w}, \boldsymbol{a}) = 0$, we have

$$\boldsymbol{K}_1\boldsymbol{a} + \boldsymbol{K}_2\boldsymbol{a} = \boldsymbol{K}_* - \lambda\boldsymbol{I}_2\boldsymbol{a},$$

and we solve for $\boldsymbol{a}^*$

$$\boldsymbol{a}^* = (\boldsymbol{K}_1 + \boldsymbol{K}_2 + \lambda\boldsymbol{I}_2)^{-1}\boldsymbol{K}_*.$$

Plugging $\boldsymbol{a}^*$ into the regularized loss function, we can get

$$\begin{aligned}
\mathcal{L}_\lambda(\boldsymbol{w}) &= \mathcal{L}_\lambda(\boldsymbol{w}, \boldsymbol{a}^*) \\
&= (\boldsymbol{a}^*)^\top (\boldsymbol{K}_* - \lambda\boldsymbol{I}_2\boldsymbol{a}^*) - 2(\boldsymbol{a}^*)^\top\boldsymbol{K}_* + \lambda(\boldsymbol{a}^*)^\top\boldsymbol{a}^* \\
&= -(\boldsymbol{a}^*)^\top\boldsymbol{K}_* \\
&= -\boldsymbol{K}_*^\top(\boldsymbol{K}_1 + \boldsymbol{K}_2 + \lambda\boldsymbol{I}_2)^{-1}\boldsymbol{K}_*
\end{aligned}$$

$\square$

Since by assumption $\boldsymbol{w}_1$ is aligned with $\boldsymbol{w}^*$, then the loss function $\mathcal{L}_\lambda(\boldsymbol{w})$ is equivalent to the following loss function $\mathcal{L}_\xi(\theta)$ by plugging in the definition of $\boldsymbol{K}_1$ and $\boldsymbol{K}_2$, where $\theta := \angle(\boldsymbol{w}^*, \boldsymbol{w}_2)$.

$$\mathcal{L}_\xi(\theta) = -\frac{c^2}{16\pi^2(d+4)^2} \begin{bmatrix} h_2(0) \\ h_2(\theta) \end{bmatrix}^\top \begin{bmatrix} \frac{c}{\pi}h_1(0) + \frac{c}{4\pi(d+4)}h_2(0) + \xi c & \frac{c}{\pi}h_1(\theta) + \frac{c}{4\pi(d+4)}h_2(\theta) \\ \frac{c}{\pi}h_1(\theta) + \frac{c}{4\pi(d+4)}h_2(\theta) & \frac{c}{\pi}h_1(0) + \frac{c}{4\pi(d+4)}h_2(0) + \xi c \end{bmatrix}^{-1} \begin{bmatrix} h_2(0) \\ h_2(\theta) \end{bmatrix},$$

where $c = \frac{2}{d+2}$ and $\lambda = \xi c$ ($\xi$ is a constant independent of $d$). When $d \to +\infty$, we have that

$$\mathcal{L}_\xi(\theta) \approx -\frac{c}{16\pi^2(d+4)^2} \frac{(\frac{1}{\pi}h_1(0) + \xi)(h_2^2(0) + h_2^2(\theta)) - 2h_2(0)h_2(\theta)\frac{1}{\pi}h_1(\theta)}{(\frac{1}{\pi}h_1(0) + \xi)^2 - (\frac{1}{\pi}h_1(\theta))^2}.$$

Noting that $h_1(0) = \pi$ and $h_2(0) = 3\pi$, we can get

$$\begin{aligned}
\mathcal{L}_\xi(\theta) &\approx -\frac{c}{16\pi^2(d+4)^2} \frac{(1+\xi)(9\pi^2 + h_2^2(\theta)) - 6h_1(\theta)h_2(\theta)}{(1+\xi)^2 - \frac{1}{\pi^2}h_1^2(\theta)} \\
&= -\frac{c}{16(d+4)^2} \frac{(9+9\xi) + \frac{1}{\pi^2}[(1+\xi)h_2^2(\theta) - 6h_1(\theta)h_2(\theta)]}{(1+\xi)^2 - \frac{1}{\pi^2}h_1^2(\theta)}.
\end{aligned}$$

Recall that we define

$$\begin{aligned}
\widetilde{\mathcal{L}}_\xi(\theta) &:= \lim_{d\to+\infty} \frac{16(d+4)^2}{c}\mathcal{L}_\xi(\theta) \\
&= -\frac{(9+9\xi) + \frac{1}{\pi^2}[(1+\xi)h_2^2(\theta) - 6h_1(\theta)h_2(\theta)]}{(1+\xi)^2 - \frac{1}{\pi^2}h_1^2(\theta)}.
\end{aligned} \tag{21}$$

Then we proceed to show the main result of $\widetilde{\mathcal{L}}_\xi(\theta)$.

**Theorem C.2.** *Consider the minimization of the limiting loss function $\widetilde{\mathcal{L}}_\xi$ defined by* (10).

1. *When $\xi \geqslant \frac{1}{2}$, $\widetilde{\mathcal{L}}_\xi(\theta)$ has a unique global minimizer at $\theta = 0$ for $\theta$ on $[0, \frac{5\pi}{6}]$.*

2. *When $\xi \leqslant \xi_0$ with some $\xi_0 < 1/2$, besides the local minimizer $\theta = 0$, there exists at least one additional local minimizer of $\widetilde{\mathcal{L}}_\xi(\theta)$ in the interval $(\frac{\pi}{4}, \frac{\pi}{2})$.*

*Proof.* Recall that

$$\begin{aligned}
\widetilde{\mathcal{L}}_\xi(\theta) &= -\frac{(9+9\xi) + \frac{1}{\pi^2}[(1+\xi)h_2^2(\theta) - 6h_1(\theta)h_2(\theta)]}{(1+\xi)^2 - \frac{1}{\pi^2}h_1^2(\theta)} \\
&= -\frac{9(1+\xi) + \frac{1}{\pi^2}[(1+\xi)B(\theta) - 6C(\theta)]}{(1+\xi)^2 - \frac{1}{\pi^2}A(\theta)},
\end{aligned}$$

where

$$A(\theta) := h_1^2(\theta), \quad B(\theta) := h_2^2(\theta), \quad C(\theta) := h_1(\theta)h_2(\theta).$$

We take the derivative of $\widetilde{\mathcal{L}}_\xi(\theta)$ with respect to $\theta$ to get

$$\mathcal{L}_\xi'(\theta) = -\frac{1}{\pi^2} \frac{\Gamma_\xi(\theta)}{[(1+\xi)^2 - \frac{1}{\pi^2}A(\theta)]^2},$$

where

$$\Gamma_\xi(\theta) := (1+\xi)^3 B'(\theta) - 6(1+\xi)^2 C'(\theta) + 9(1+\xi)A'(\theta) + \frac{1+\xi}{\pi^2}(A'(\theta)B(\theta) - A(\theta)B'(\theta))$$

$$+ \frac{6}{\pi^2}(A(\theta)C'(\theta) - A'(\theta)C(\theta)).$$

Recall that

$$h_1(\theta) = \sin\theta\cos\theta + (\pi - \theta)\cos^2\theta,$$
$$h_1'(\theta) = -\sin^2\theta - 2(\pi - \theta)\sin\theta\cos\theta,$$
$$h_2(\theta) = 3\sin\theta\cos\theta + (\pi - \theta)(1 + 2\cos^2\theta),$$
$$h_2'(\theta) = -4\sin^2\theta - 4(\pi - \theta)\sin\theta\cos\theta.$$

By calculation, we have

$$A(\theta) = \sin^2\theta\cos^2\theta + 2(\pi - \theta)\sin\theta\cos^3\theta + (\pi - \theta)^2\cos^4\theta$$
$$B(\theta) = 9\sin^2\theta\cos^2\theta + 6(\pi - \theta)\sin\theta\cos\theta(1 + 2\cos^2\theta) + (\pi - \theta)^2(1 + 2\cos^2\theta)^2$$
$$C(\theta) = 3\sin^2\theta\cos^2\theta + (\pi - \theta)\sin\theta\cos\theta(1 + 5\cos^2\theta) + (\pi - \theta)^2(\cos^2\theta + 2\cos^4\theta)$$
$$A'(\theta) = -2\sin^3\theta\cos\theta - 6(\pi - \theta)\sin^2\theta\cos^2\theta - 4(\pi - \theta)^2\sin\theta\cos^3\theta$$
$$B'(\theta) = -24\sin^3\theta\cos\theta - 8(\pi - \theta)\sin^2\theta(1 + 5\cos^2\theta) - 8(\pi - \theta)^2\sin\theta\cos\theta(1 + 2\cos^2\theta)$$
$$C'(\theta) = -7\sin^3\theta\cos\theta - (\pi - \theta)\sin^2\theta(1 + 16\cos^2\theta) - 2(\pi - \theta)^2\sin\theta\cos\theta(1 + 4\cos^2\theta)$$
$$A'(\theta)B(\theta) - A(\theta)B'(\theta) = 6\sin^5\theta\cos^3\theta + 2(\pi - \theta)\sin^4\theta\cos^2\theta(-2 + 5\cos^2\theta)$$
$$+ 2(\pi - \theta)^2\sin^3\theta\cos\theta(-1 - 10\cos^2\theta + 2\cos^4\theta)$$
$$+ 2(\pi - \theta)^3\sin^2\theta\cos^2\theta(-3 - 12\cos^2\theta)$$
$$+ 4(\pi - \theta)^4\sin\theta\cos^3\theta(-1 - 2\cos^2\theta)$$
$$A(\theta)C'(\theta) - A'(\theta)C(\theta) = -\sin^5\theta\cos^3\theta + (\pi - \theta)\sin^4\theta\cos^2\theta(1 - 2\cos^2\theta)$$
$$+ (\pi - \theta)^2\sin^3\theta\cos^3\theta(4 - \cos^2\theta)$$
$$+ 5(\pi - \theta)^3\sin^2\theta\cos^4\theta$$
$$+ 2(\pi - \theta)^4\sin\theta\cos^5\theta$$

Hence, by letting $\bar{\xi} := 1 + \xi$, we have

$$\Gamma_{\bar{\xi}}(\theta) = (-24\bar{\xi}^3 + 42\bar{\xi}^2 - 18\bar{\xi})\sin^3\theta\cos\theta + (\pi - \theta)\sin^2\theta\left(6\bar{\xi}^2 - 8\bar{\xi}^3 + (-40\bar{\xi}^3 + 96\bar{\xi}^2 - 54\bar{\xi})\cos^2\theta\right)$$

$$+ (\pi - \theta)^2\sin\theta\cos\theta\left(12\bar{\xi}^2 - 8\bar{\xi}^3 + (-16\bar{\xi}^3 + 48\bar{\xi}^2 - 36\bar{\xi})\cos^2\theta\right)$$

$$+ \frac{1}{\pi^2}\Phi_{\bar{\xi}}(\theta),$$

where

$$\Phi_{\bar{\xi}}(\theta) := 6(\bar{\xi} - 1)\sin^5\theta\cos^3\theta + 2(\pi - \theta)\sin^4\theta\cos^2\theta(-2\bar{\xi} + 3 + (5\bar{\xi} - 6)\cos^2\theta)$$

$$+ 2(\pi - \theta)^2\sin^3\theta\cos\theta(-\bar{\xi} + (12 - 10\bar{\xi})\cos^2\theta + (2\bar{\xi} - 3)\cos^4\theta)$$

$$+ 2(\pi - \theta)^3\sin^2\theta\cos^2\theta(-3\bar{\xi} + (15 - 12\bar{\xi})\cos^2\theta)$$

$$+ 4(\pi - \theta)^4\sin\theta\cos^3\theta(-\bar{\xi} + (3 - 2\bar{\xi})\cos^2\theta).$$

We also note that $\Gamma_\xi(\theta)$ can be written as

$$\Gamma_\xi(\theta) = \gamma_0(\theta) + \xi\gamma_1(\theta) + \xi^2\gamma_2(\theta) + \xi^3\gamma_3(\theta),$$

where

$$\gamma_0(\theta) := -2(\pi - \theta)\sin^4\theta + 4(\pi - \theta)^2\sin^3\theta\cos\theta$$
$$+ \frac{1}{\pi^2}\left[2(\pi-\theta)\sin^6\theta\cos^2\theta - 2(\pi-\theta)^2\sin^7\theta\cos\theta - 6(\pi-\theta)^3\sin^4\theta\cos^2\theta - 4(\pi-\theta)^4\sin^3\theta\cos^3\theta\right],$$

$$\gamma_1(\theta) := -6\sin^3\theta\cos\theta + (\pi-\theta)\sin^2\theta\left(-12 + 18\cos^2\theta\right) + 12(\pi-\theta)^2\sin\theta\cos^3\theta + \frac{1}{\pi^2}\tilde{\gamma}_1(\theta),$$

where

$$\tilde{\gamma}_1(\theta) := 6\sin^5\theta\cos^3\theta + 2(\pi-\theta)\sin^4\theta\cos^2\theta(-2 + 5\cos^2\theta)$$
$$+ 2(\pi-\theta)^2\sin^3\theta\cos\theta(-1 - 10\cos^2\theta + 2\cos^4\theta)$$
$$- 6(\pi-\theta)^3\sin^2\theta\cos^2\theta(1 + 4\cos^2\theta)$$
$$- 4(\pi-\theta)^4\sin\theta\cos^3\theta(1 + 2\cos^2\theta),$$

$$\gamma_2(\theta) := -30\sin^3\theta\cos\theta - 6(\pi-\theta)\sin^2\theta\left(3 + 4\cos^2\theta\right) - 12(\pi-\theta)^2\sin\theta\cos\theta,$$
$$\gamma_3(\theta) := -24\sin^3\theta\cos\theta - 8(\pi-\theta)\sin^2\theta\left(1 + 5\cos^2\theta\right) - 8(\pi-\theta)^2\sin\theta\cos\theta(1 + 2\cos^2\theta).$$

**Claim 1: For any $\xi > 0$, $\widetilde{\mathcal{L}}_\xi(\theta)$ is increasing on $(\frac{\pi}{2}, \frac{5\pi}{6}]$.**
We want to show that $\Gamma_\xi(\theta) < 0$ for $\theta \in (\frac{\pi}{2}, \frac{5\pi}{6}]$. To show that, we will show $\gamma_0(\theta) < 0$, $\gamma_1(\theta) < 0$, $\gamma_2(\theta) < 0$ and $\gamma_3(\theta) < 0$ respectively.
To show $\gamma_0(\theta) < 0$ on $(\frac{\pi}{2}, \pi)$, we observe that

$$\frac{2}{\pi^2}(\pi-\theta)\sin^6\theta\cos^2\theta < \frac{2}{\pi^2}(\pi-\theta)\sin^4\theta,$$
$$-\frac{2}{\pi^2}(\pi-\theta)^2\sin^7\theta\cos\theta < -\frac{2}{\pi^2}(\pi-\theta)^2\sin^3\theta\cos\theta,$$
$$-\frac{4}{\pi^2}(\pi-\theta)^4\sin^3\theta\cos^3\theta < -\frac{4}{\pi^2}(\pi-\theta)^2\frac{\pi^2}{4}\sin^3\theta\cos\theta = -(\pi-\theta)^2\sin^3\theta\cos\theta,$$

by using $\sin^2\theta < 1$, $\cos^2\theta < 1$ and $(\pi-\theta)^2 < (\frac{\pi}{2})^2$. Then we can get

$$\gamma_0(\theta) < -\left(2 - \frac{2}{\pi^2}\right)(\pi-\theta)\sin^4\theta + \left(3 - \frac{2}{\pi^2}\right)(\pi-\theta)^2\sin^3\theta\cos\theta < 0.$$

Next, we show that $\gamma_1(\theta) < 0$ on $(\frac{\pi}{2}, \frac{5\pi}{6}]$. First, we note that

$$6\sin^5\theta\cos^3\theta \leqslant 0 \text{ (the first term in } \tilde{\gamma}_1(\theta)),$$
$$2(\pi-\theta)\sin^4\theta\cos^2\theta(-2 + 5cos^2\theta) \leqslant \frac{7}{2}(\pi-\theta)\sin^2\theta\cos^2\theta \text{ (the second term in } \tilde{\gamma}_1(\theta))$$

by using $\cos^2\theta \leqslant \frac{3}{4}$ and $\sin^2\theta \leqslant 1$. Then we focus on the sum of the forth term and the fifth term in $\tilde{\gamma}_1(\theta)$ and let $t := \pi - \theta$. For $\theta \in (\frac{\pi}{2}, \pi)$, $t \in (0, \frac{\pi}{2})$, then we can get

$$H_1(t) := -6t^3\sin^2 t\cos^2 t(1 + 4\cos^2 t) + 4t^4\sin t\cos^3 t(1 + 2\cos^2 t)$$
$$= 2t^3\sin t\cos^2 t\left(-3\sin t(1 + 4\cos^2 t) + 2t\cos t(1 + 2\cos^2 t)\right)$$
$$\leqslant 2t^3\sin t\cos^2 t\left(-3\sin t(1 + 4\cos^2 t) + 2\sin t(1 + 2\cos^2 t)\right)$$
$$= 2t^3\sin t\cos^2 t\left(-\sin t - 8\sin t\cos^2 t\right)$$
$$< 0,$$

where the first inequality comes from the standard inequality $\tan x \geqslant x$ on $(0, \frac{\pi}{2})$. Combining the above inequalities, we can get on $(\frac{\pi}{2}, \frac{5\pi}{6}]$,

$$\gamma_1(\theta) < -6\sin^3\theta\cos\theta - 12(\pi-\theta)\sin^2\theta + \left(18 + \frac{7}{2\pi^2}\right)(\pi-\theta)\sin^2\theta\cos^2\theta + 12(\pi-\theta)^2\sin\theta\cos^3\theta$$
$$+ \frac{1}{\pi^2}\left[2(\pi-\theta)^2\sin^3\theta\cos\theta(-1 - 10\cos^2\theta + 2\cos^4\theta)\right]$$
$$< -6\sin^3\theta\cos\theta - 12(\pi-\theta)\sin^2\theta + 19(\pi-\theta)\sin^2\theta\cos^2\theta + 12(\pi-\theta)^2\sin\theta\cos^3\theta$$
$$+ \frac{1}{\pi^2}\left[2(\pi-\theta)^2\sin^3\theta\cos\theta(-1 - 10\cos^2\theta + 2\cos^4\theta)\right].$$

By letting $t := \pi - \theta$ and using properties $\sin\theta = \sin t > 0$ and $\cos\theta = -\cos t < 0$ when $\theta \in (\frac{\pi}{2}, \frac{5\pi}{6}]$, $t \in [\frac{\pi}{6}, \frac{\pi}{2})$. The above inequality can be written as

$$\gamma_1(\theta) < \sin t J_1(t),$$

where

$$J_1(t) := 6\sin^2 t \cos t - 12t \sin t + 19t \sin t \cos^2 t - 12t^2 \cos^3 t$$
$$+ \frac{1}{\pi^2}\left[2t^2 \sin^2 t \cos t(1 + 10\cos^2 t - 2\cos^4 t)\right].$$

Since $\cos^2 t \leqslant \frac{3}{4}$ and $\sin^2 t \geqslant \frac{1}{4}$ on $[\frac{\pi}{6}, \frac{\pi}{2})$, we have that

$$J_1(t) \leqslant 6\sin^2 t \cos t - 12t \sin t + 19t \sin t \cos^2 t - 12t^2 \cos^3 t + \frac{1}{\pi^2}\left[17t^2 \sin^2 t \cos t - 4t^2 \sin^2 t \cos^5 t\right]$$

$$\leqslant 6\sin^2 t \cos t - 12t \sin t + 19t \sin t \cos^2 t - 12t^2 \cos^3 t + \frac{1}{\pi^2}\left[17t^2 \sin^2 t \cos t - t^2 \cos^5 t\right]$$

$$< 6\sin^2 t \cos t - 12t \sin t + 19t \sin t \cos^2 t - 12t^2 \cos^3 t + \frac{17}{\pi^2}t^2 \sin^2 t \cos t$$

$$< 6\sin^2 t \cos t - 12t \sin t + 19t \sin t \cos^2 t - 12t^2 \cos^3 t + 2t^2 \sin^2 t \cos t.$$

We divide the interval $[\frac{\pi}{6}, \frac{\pi}{2})$ into two sub-intervals $[\frac{\pi}{6}, 1.4]$ and $[1.4, \frac{\pi}{2})$. On $[1.4, \frac{\pi}{2})$, by using inequalities $\sin t \in [\sin(1.4), 1)$, $\cos t \in (0, \cos(1.4)]$ and $t \in [1.4, \frac{\pi}{2})$, we can get

$$J_1(t) < 6\cos(1.4) - 12 \cdot 1.4\sin(1.4) + 19\frac{\pi}{2}\cos^2(1.4) + 2\frac{\pi^2}{4}\cos(1.4) \approx -13.83 < 0$$

On $[\frac{\pi}{6}, 1.4]$, we take the first derivative of $J_1(t)$ to get

$$J_1'(t) = 19\sin t - 37\sin^3 t - 46t\cos t + 29t\cos^3 t + 40t^2 \sin t \cos^2 t - 2t^2 \sin^3 t$$

$$\leqslant 19\sin t - 37\sin^3 t - \frac{97}{4}t\cos t + 40t^2 \sin t \cos^2 t - 2t^2 \sin^3 t$$

$$= 19\sin t + 40t^2 \sin t - 37\sin^3 t - 42t^2 \sin^3 t - \frac{97}{4}t\cos t,$$

where the inequality comes from the fact that $\cos t \leqslant \frac{\sqrt{3}}{2}$ on $[\frac{\pi}{6}, 1.4]$. By using following standard Taylor polynomial bounds

$$\sin t \geqslant t - \frac{t^3}{6}, \quad \sin t \leqslant t - \frac{t^3}{6} + \frac{t^5}{120}, \quad \cos t \geqslant 1 - \frac{t^2}{2},$$

we can get

$$J_1'(t) \leqslant P_1(t) := \frac{7}{36}t^{11} - \frac{719}{216}t^9 + \frac{73}{4}t^7 - \frac{3601}{120}t^5 + \frac{287}{24}t^3 - \frac{21}{4}t.$$

We can check that $P_1(t) < 0$ on $[\frac{\pi}{6}, 1.4]$, hence, $J_1'(t) < 0$ on $[\frac{\pi}{6}, 1.4]$, which implies that $J_1(t)$ is decreasing on $[\frac{\pi}{6}, 1.4]$ and hence $J_1(t) \leqslant J_1(\frac{\pi}{6}) < 0$ on $[\frac{\pi}{6}, 1.4]$. Then we can conclude that $\gamma_1(\theta) < 0$ on $(\frac{\pi}{2}, \frac{5\pi}{6}]$.

To show $\gamma_2(\theta) < 0$, we let $t := \pi - \theta$, then $t \in (0, \frac{\pi}{2})$. We use properties $\sin\theta = \sin t > 0$ and $\cos\theta = -\cos t < 0$, then we have

$$\gamma_2(\theta) = -6\sin t J_2(t),$$

where $J_2(t) := t\sin t(3 + 4\cos^2 t) - \cos t(2t^2 + 5\sin^2 t)$. Next, we just need to show that $J_2(t) > 0$. Take the first derivative and second derivative of $J_2(t)$, we can get

$$J_2'(t) = 12\sin t + 3t\cos t - 4\sin^3 t + 2t^2 \sin t - 12t\sin^2 t \cos t - 15\sin t \cos^2 t,$$

$$J_2''(t) = -23t\sin t + 36t\sin^3 t + 21\sin^2 t \cos t + 2t^2 \cos t.$$

By using following standard Taylor polynomial bounds on $[0, \frac{\pi}{2}]$ for $J_2''(t)$

$$\sin t \geqslant t - \frac{t^3}{6}, \quad \sin t \leqslant t - \frac{t^3}{6} + \frac{t^5}{120}, \quad \cos t \geqslant 1 - \frac{t^2}{2},$$

we can show that

$$J_2''(t) \geqslant P_2(t) := t^4 R_2(t^2),$$

where $R_2(x) = -\frac{1}{6}x^3 + \frac{65}{24}x^2 - \frac{1693}{120}x + \frac{64}{3}$. By analyzing $R_2'(x)$, it is easy to check that $R_2(x)$ is decreasing on $[0, (\frac{\pi}{2})^2]$. Hence, $R_2(t^2) \geqslant R_2((\frac{\pi}{2})^2) > 0$, which means $J_2''(t) > 0$ on $[0, \frac{\pi}{2}]$. Therefore, $J_2'(t) > J_2'(0) = 0$ on $(0, \frac{\pi}{2})$, which is equivalent to $J_2(t) > J_2(0) = 0$ on $(0, \frac{\pi}{2})$. At last, we can conclude that $\gamma_2(\theta) < 0$ on $(\frac{\pi}{2}, \pi)$.

To show $g_3(\theta) < 0$, similarly, we let $t := \pi - \theta$, then $t \in (0, \frac{\pi}{2})$. We use properties $\sin \theta = \sin t > 0$ and $\cos \theta = -\cos t < 0$, then we have

$$\gamma_3(\theta) = 8 \sin t J_3(t),$$

where $J_3(t) := 3 \sin^2 t \cos t - t \sin t(1 + 5\cos^2 t) + t^2 \cos t(1 + 2\cos^2 t)$. Take the first derivative of $J_3(t)$, we can get

$$J_3'(t) = \sin t H_3(t),$$

where

$$H_3(t) := (6t^2 - 4)\sin^2 t - 7t^2 + 11t \sin t \cos t.$$

By using following standard Taylor polynomial bounds on $[0, \frac{\pi}{2}]$ for $H_3(t)$

$$\sin t \geqslant t - \frac{t^3}{6}, \quad \sin t \leqslant t - \frac{t^3}{6} + \frac{t^5}{120}, \quad \cos t \geqslant 1 - \frac{t^2}{2},$$

we can show that

$$H_3(t) \leqslant P_3(t) := t^6 R_3(t^2),$$

where $R_3(x) = \frac{1}{2400}x^3 - \frac{37}{2880}x^2 + \frac{13}{90}x - \frac{87}{135}$. By analyzing $R_3'(x)$, it is easy to check that $R_3(x)$ is increasing on $[0, (\frac{\pi}{2})^2]$. Hence, $R_3(t^2) \leqslant R_3((\frac{\pi}{2})^2) < 0$, which means that $H_3(t) < 0$ on $(0, \frac{\pi}{2})$ and further implies that $J_3(t)$ is decreasing on $(0, \frac{\pi}{2})$. Therefore, $J_3(t) < J_3(0) = 0$ on $(0, \frac{\pi}{2})$, which further implies that $\gamma_3(\theta) < 0$ on $(\frac{\pi}{2}, \pi)$.

In summary, we have shown that $\Gamma_\xi(\theta) < 0$ on $(\frac{\pi}{2}, \frac{5\pi}{6}]$ for any $\xi > 0$, which means that $\widetilde{\mathcal{L}}_\xi(\theta)$ is increasing on $(\frac{\pi}{2}, \frac{5\pi}{6}]$ for any $\xi > 0$.

**Claim 2:** $\theta = 0$ **is a local minimizer for any** $\xi > 0$.
First, we observe that $\widetilde{\mathcal{L}}_\xi'(0) = 0$. Then We take the second derivative of $\widetilde{\mathcal{L}}_\xi(\theta)$ with respect to $\theta$ to get

$$\widetilde{\mathcal{L}}_\xi''(\theta) = -\frac{1}{\pi^2} \frac{\Gamma_\xi'(\theta)[(1+\xi)^2 - \frac{1}{\pi^2}A(\theta)]^2 + \Gamma_\xi(\theta)\frac{2}{\pi^2}[(1+\xi)^2 - \frac{1}{\pi^2}A(\theta)]A'(\theta)}{[(1+\xi)^2 - \frac{1}{\pi^2}A(\theta)]^4},$$

where

$$\Gamma_\xi(\theta) = (1+\xi)^3 B'(\theta) - 6(1+\xi)^2 C'(\theta) + 9(1+\xi)A'(\theta) + \frac{1+\xi}{\pi^2}(A'(\theta)B(\theta) - A(\theta)B'(\theta))$$

$$+ \frac{6}{\pi^2}(A(\theta)C'(\theta) - A'(\theta)C(\theta)),$$

$$\Gamma_\xi'(\theta) = (1+\xi)^3 B''(\theta) - 6(1+\xi)^2 C''(\theta) + 9(1+\xi)A''(\theta) + \frac{1+\xi}{\pi^2}(A''(\theta)B(\theta) - A(\theta)B''(\theta))$$

$$+ \frac{6}{\pi^2}(A(\theta)C''(\theta) - A''(\theta)C(\theta)).$$

By calculation, we have

$A''(\theta) = 2\sin^4 \theta - 4(\pi - \theta)\sin\theta\cos\theta(4\cos^2\theta - 3) - 4(\pi - \theta)^2\cos^2\theta(4\cos^2\theta - 3)$

$B''(\theta) = 32\sin^4\theta - 24\sin^2\theta\cos^2\theta - 16(\pi - \theta)\sin\theta\cos\theta(8\cos^2\theta - 5) - 8(\pi - \theta)^2(-1 - 4\cos^2\theta + 8\cos^4\theta)$

$C''(\theta) = 8\sin^4\theta - 4\sin^2\theta\cos^2\theta - 2(\pi - \theta)\sin\theta\cos\theta(-17 + 24\cos^2\theta) - 2(\pi - \theta)^2(-1 - 10\cos^2\theta + 16\cos^4\theta)$

and

$$\widetilde{\mathcal{L}}_\xi''(0) = -\frac{-24\xi^3 - 12\xi^2 - 48\xi - 48}{(\xi^2 + 2\xi)^2} > 0,$$

for any $\xi > 0$. Therefore, $\theta = 0$ is a local minimizer for any $\xi > 0$.

**Claim 3: When** $\xi \geqslant \frac{1}{2}$, $\widetilde{\mathcal{L}}_\xi(\theta)$ **is increasing on** $(0, \frac{\pi}{2})$.
Recall that with $\bar{\xi} := \xi + 1$,

$\Gamma_{\bar{\xi}}(\theta) = (-24\bar{\xi}^3 + 42\bar{\xi}^2 - 18\bar{\xi})\sin^3\theta\cos\theta + (\pi - \theta)\sin^2\theta\left(6\bar{\xi}^2 - 8\bar{\xi}^3 + (-40\bar{\xi}^3 + 96\bar{\xi}^2 - 54\bar{\xi})\cos^2\theta\right)$

$$+ (\pi - \theta)^2 \sin\theta\cos\theta\left(12\bar{\xi}^2 - 8\bar{\xi}^3 + (-16\bar{\xi}^3 + 48\bar{\xi}^2 - 36\bar{\xi})\cos^2\theta\right)$$

$$+ \frac{1}{\pi^2}\Phi_{\bar{\xi}}(\theta),$$

where

$\Phi_{\bar{\xi}}(\theta) = 6(\bar{\xi} - 1)\sin^5\theta\cos^3\theta + 2(\pi - \theta)\sin^4\theta\cos^2\theta(-2\bar{\xi} + 3 + (5\bar{\xi} - 6)\cos^2\theta)$

$$+ 2(\pi - \theta)^2\sin^3\theta\cos\theta(-\bar{\xi} + (12 - 10\bar{\xi})\cos^2\theta + (2\bar{\xi} - 3)\cos^4\theta)$$

$$+ 2(\pi - \theta)^3\sin^2\theta\cos^2\theta(-3\bar{\xi} + (15 - 12\bar{\xi})\cos^2\theta)$$

$$+ 4(\pi - \theta)^4\sin\theta\cos^3\theta(-\bar{\xi} + (3 - 2\bar{\xi})\cos^2\theta).$$

Note that when $\xi \geqslant \frac{1}{2}, \bar{\xi} \geqslant \frac{3}{2}$ and we have following inequality

$$(-24\bar{\xi}^3 + 42\bar{\xi}^2 - 18\bar{\xi})\sin^3\theta\cos\theta + \frac{6}{\pi^2}(\bar{\xi}-1)\sin^5\theta\cos^3\theta < (-24\bar{\xi}^3 + 42\bar{\xi}^2 - 12\bar{\xi} - 6)\sin^3\theta\cos\theta < 0,$$

$$(\pi-\theta)^2\sin\theta\cos\theta\left(12\bar{\xi}^2 - 8\bar{\xi}^3 + (-16\bar{\xi}^3 + 48\bar{\xi}^2 - 36\bar{\xi})\cos^2\theta\right) < 0$$

Also, for other terms in $\Phi_{\bar{\xi}}(\theta)$, we have that

$$2(\pi-\theta)\sin^4\theta\cos^2\theta(-2\bar{\xi} + 3 + (5\bar{\xi}-6)\cos^2\theta) < 2(\pi-\theta)\sin^2\theta(3\bar{\xi}-3),$$

$$2(\pi-\theta)^2\sin^3\theta\cos\theta(-\bar{\xi} + (12-10\bar{\xi})\cos^2\theta + (2\bar{\xi}-3)\cos^4\theta) < 2(\pi-\theta)^2\sin^3\theta\cos\theta(-\bar{\xi} + (9-8\bar{\xi})\cos^2\theta) < 0,$$

$$2(\pi-\theta)^3\sin^2\theta\cos^2\theta(-3\bar{\xi} + (15-12\bar{\xi})\cos^2\theta) < 0,$$

$$4(\pi-\theta)^4\sin\theta\cos^3\theta(-\bar{\xi} + (3-2\bar{\xi})\cos^2\theta) < 0.$$

Then we combine the second term of $\Gamma_{\bar{\xi}}(\theta)$ and the first inequality above to get

$$(\pi-\theta)\sin^2\theta\left(6\bar{\xi}^2 - 8\bar{\xi}^3 + (-40\bar{\xi}^3 + 96\bar{\xi}^2 - 54\bar{\xi})\cos^2\theta\right) + \frac{2}{\pi^2}(\pi-\theta)\sin^4\theta\cos^2\theta(-2\bar{\xi} + 3 + (5\bar{\xi}-6)\cos^2\theta)$$

$$<(\pi-\theta)\sin^2\theta\left(6\bar{\xi}^2 - 8\bar{\xi}^3 + 6\bar{\xi} - 6 + (-40\bar{\xi}^3 + 96\bar{\xi}^2 - 54\bar{\xi})\cos^2\theta\right) < 0$$

Therefore, we can get $\Gamma_{\bar{\xi}}(\theta) < 0$ when $\bar{\xi} \geqslant \frac{3}{2}$ and $\theta \in (0, \frac{\pi}{2})$. Hence, $\widetilde{\mathcal{L}}'_\xi(\theta)$ is positive, which implies that $\widetilde{\mathcal{L}}_\xi(\theta)$ is increasing on $(0, \frac{\pi}{2})$ when $\bar{\xi} \geqslant \frac{3}{2}$.

**Claim 4: When $\xi \leqslant 0.13$, there is at least one additional local minimizer on $(\frac{\pi}{4}, \frac{\pi}{2})$.**
We first note that
$$\Gamma_{\bar{\xi}}(\frac{\pi}{2}) = \frac{\pi}{2}(6\bar{\xi}^2 - 8\bar{\xi}^3) < 0 \Rightarrow \mathcal{L}'_\xi(\frac{\pi}{2}) > 0.$$

and

$$\gamma_0(\frac{\pi}{4}) \approx 1.7777, \quad \gamma_1(\frac{\pi}{4}) \approx -2.0681, \quad \gamma_2(\frac{\pi}{4}) \approx -76.1528, \quad \gamma_3(\frac{\pi}{4}) \approx -58.8614,$$

$$\Gamma_{\bar{\xi}}(\frac{\pi}{4}) = \gamma_0(\frac{\pi}{4}) + \xi\gamma_1(\frac{\pi}{4}) + \xi^2\gamma_2(\frac{\pi}{4}) + \xi^3\gamma_3(\frac{\pi}{4}).$$

By calculation, we can get when $\xi \leqslant 0.13$,

$$\Gamma_{\bar{\xi}}(\theta)(\frac{\pi}{4}) \geqslant \gamma_0(\frac{\pi}{4}) + 0.13\cdot\gamma_1(\frac{\pi}{4}) + 0.13^2\cdot\gamma_2(\frac{\pi}{4}) + 0.13^3\cdot\gamma_3(\frac{\pi}{4}) > 0 \Rightarrow \mathcal{L}'_\xi(\frac{\pi}{4}) < 0.$$

Hence, $\mathcal{L}_\xi(\theta)$ has at least one local minimizer in the interval $(\frac{\pi}{4}, \frac{\pi}{2})$.

Finally, by combining above claims, we complete the proof of the theorem.

$\square$

# D   PROOF OF LEMMA 2.6 AND THEOREM 2.7

In this section, we present the proof of Lemma 2.6 and Theorem 2.7. For completeness of the section, we include Lemma 2.6 and Theorem 2.7 again.

**Lemma D.1.** *For any fixed $\boldsymbol{w}$, the minimizer $\boldsymbol{a}^*$ of the regularized loss function (11) has a closed form $\boldsymbol{a}^* = \left(\boldsymbol{K}_1 + \boldsymbol{K}_2 + \begin{bmatrix} 0 & 0 \\ 0 & \lambda \end{bmatrix}\right)^{-1}\boldsymbol{K}_*$ and $\mathcal{L}_\lambda(\boldsymbol{w}) = -\boldsymbol{K}_*^\top\left(\boldsymbol{K}_1 + \boldsymbol{K}_2 + \begin{bmatrix} 0 & 0 \\ 0 & \lambda \end{bmatrix}\right)^{-1}\boldsymbol{K}_*.$*

*Proof.* Recall that
$$\mathcal{L}_\lambda(\boldsymbol{w}, \boldsymbol{a}) = \boldsymbol{a}^\top\boldsymbol{K}_1\boldsymbol{a} + \boldsymbol{a}^\top\boldsymbol{K}_2\boldsymbol{a} - 2\boldsymbol{a}^\top\boldsymbol{K}_* + \lambda a_2^2.$$

To find the minimizer $\boldsymbol{a}^*$, we take the gradient of $\mathcal{L}_\lambda(\boldsymbol{w}, \boldsymbol{a})$ with respect to $\boldsymbol{a}$ to get

$$\nabla_a L_\lambda(\boldsymbol{w}, \boldsymbol{a}) = 2\boldsymbol{K}_1\boldsymbol{a} + 2\boldsymbol{K}_2\boldsymbol{a} - 2\boldsymbol{K}_* + 2\begin{bmatrix} 0 \\ \lambda a_2 \end{bmatrix}.$$

By letting $\nabla_a\mathcal{L}_\lambda(\boldsymbol{w}, \boldsymbol{a}) = 0$, we have

$$\boldsymbol{K}_1\boldsymbol{a} + \boldsymbol{K}_2\boldsymbol{a} = \boldsymbol{K}_* - \begin{bmatrix} 0 \\ \lambda a_2 \end{bmatrix},$$

and we solve for $a^*$

$$a^* = \left( K_1 + K_2 + \begin{bmatrix} 0 & 0 \\ 0 & \lambda \end{bmatrix} \right)^{-1} K_*.$$

Plugging $a^*$ into the regularized loss function, we can get

$$
\begin{aligned}
\mathcal{L}_\lambda(w) &= \mathcal{L}_\lambda(w, a^*) \\
&= (a^*)^\top \left( K_* - \begin{bmatrix} 0 \\ \lambda a_2^* \end{bmatrix} \right) - 2(a^*)^\top K_* + \lambda(a_2^*)^2 \\
&= -(a^*)^\top K_* \\
&= -K_*^\top \left( K_1 + K_2 + \begin{bmatrix} 0 & 0 \\ 0 & \lambda \end{bmatrix} \right)^{-1} K_*
\end{aligned}
\tag{22}
$$

$\square$

Since by assumption $w_1$ is aligned with $w^*$, then the loss function $\mathcal{L}_\lambda(w)$ is equivalent to the following loss function $\mathcal{L}_\xi(\theta)$ by plugging in the definition of $K_1$ and $K_2$, where $\theta := \angle(w^*, w_2)$.

$$\mathcal{L}_\xi(\theta) = -\frac{c^2}{16\pi^2(d+4)^2} \begin{bmatrix} h_2(0) \\ h_2(\theta) \end{bmatrix}^\top \begin{bmatrix} \frac{c}{\pi}h_1(0) + \frac{c}{4\pi(d+4)}h_2(0) & \frac{c}{\pi}h_1(\theta) + \frac{c}{4\pi(d+4)}h_2(\theta) \\ \frac{c}{\pi}h_1(\theta) + \frac{c}{4\pi(d+4)}h_2(\theta) & \frac{c}{\pi}h_1(0) + \frac{c}{4\pi(d+4)}h_2(0) + \xi c \end{bmatrix}^{-1} \begin{bmatrix} h_2(0) \\ h_2(\theta) \end{bmatrix},$$

where $c = \frac{2}{d+2}$ and $\lambda = \xi c$ ($\xi$ is a constant independent of $d$). When $d \to +\infty$, we have that

$$\mathcal{L}_\xi(\theta) \approx -\frac{c}{16\pi^2(d+4)^2} \frac{h_2^2(0)(\frac{1}{\pi}h_1(0) + \xi) + h_2^2(\theta)\frac{1}{\pi}h_1(0) - 2h_2(0)h_2(\theta)\frac{1}{\pi}h_1(\theta)}{\frac{1}{\pi}h_1(0)(\frac{1}{\pi}h_1(0) + \xi) - (\frac{1}{\pi}h_1(\theta))^2}.$$

Noting that $h_1(0) = \pi$ and $h_2(0) = 3\pi$, we can get

$$\mathcal{L}_\xi(\theta) \approx -\frac{c}{16(d+4)^2} \frac{(9+9\xi) + \frac{1}{\pi^2}[h_2^2(\theta) - 6h_1(\theta)h_2(\theta)]}{1 + \xi - \frac{1}{\pi^2}h_1^2(\theta)}.$$

Recall that we define

$$
\begin{aligned}
\bar{\mathcal{L}}_\xi(\theta) &:= \lim_{d \to +\infty} \frac{16(d+4)^2}{c} \mathcal{L}_\xi(\theta) \\
&= -\frac{(9+9\xi) + \frac{1}{\pi^2}[h_2^2(\theta) - 6h_1(\theta)h_2(\theta)]}{1 + \xi - \frac{1}{\pi^2}h_1^2(\theta)}.
\end{aligned}
$$

Then we proceed to show the main result of $\bar{\mathcal{L}}_\xi(\theta)$.

**Theorem D.2.** *Consider the minimization of the limiting loss function $\bar{\mathcal{L}}_\xi$ defined by (12), there exists a **unique** global minimizer $\theta^* \in (\frac{\pi}{3}, \frac{\pi}{2})$ for $\bar{\mathcal{L}}_\xi(\theta)$ for any $\xi > 0$.*

*Proof.* Recall that

$$\bar{\mathcal{L}}_\xi(\theta) = -\frac{(9+9\xi) + \frac{1}{\pi^2}[h_2^2(\theta) - 6h_1(\theta)h_2(\theta)]}{1 + \xi - \frac{1}{\pi^2}h_1^2(\theta)}.$$

We denote

$$
\begin{aligned}
E(\theta) &:= h_2^2(\theta) - 6h_1(\theta)h_2(\theta) = h_2(\theta)(h_2(\theta) - 6h_1(\theta)), \\
&= [3\sin\theta\cos\theta + (\pi-\theta)(1+2\cos^2\theta)][-3\sin\theta\cos\theta + (\pi-\theta)(1-4\cos^2\theta)] \\
F(\theta) &:= h_1^2(\theta) = [\sin\theta\cos\theta + (\pi-\theta)\cos^2\theta]^2.
\end{aligned}
$$

Then we take the derivative of $\bar{\mathcal{L}}_\xi(\theta)$ to get

$$\bar{\mathcal{L}}_\xi'(\theta) = -\frac{1}{\pi^2} \frac{(1+\xi)(E'(\theta) + 9F'(\theta)) + \frac{1}{\pi^2}(E(\theta)F'(\theta) - E'(\theta)F(\theta))}{[1 + \xi - \frac{1}{\pi^2}F(\theta)]^2}.$$

Recall that

$$h_1(\theta) = \sin\theta\cos\theta + (\pi - \theta)\cos^2\theta,$$

$$h_1'(\theta) = -\sin^2\theta - 2(\pi - \theta)\sin\theta\cos\theta,$$

$$h_2(\theta) = 3\sin\theta\cos\theta + (\pi - \theta)(1 + 2\cos^2\theta),$$

$$h_2'(\theta) = -4\sin^2\theta - 4(\pi - \theta)\sin\theta\cos\theta.$$

Then by calculation, we have the following equations.

$$F'(\theta) = -2\sin^3\theta\cos\theta - 6(\pi - \theta)\sin^2\theta\cos^2\theta - 4(\pi - \theta)^2\sin\theta\cos^3\theta,$$

$$E'(\theta) = 18\sin^3\theta\cos\theta + 54(\pi - \theta)\sin^2\theta\cos^2\theta - 2(\pi - \theta)\sin^4\theta + 4(\pi - \theta)^2\sin\theta\cos\theta(1 + 8\cos^2\theta),$$

$$E'(\theta) + 9F'(\theta) = -2(\pi - \theta)\sin^4\theta + 4(\pi - \theta)^2\sin^3\theta\cos\theta,$$

$$F(\theta) = h_1^2(\theta) = \sin^2\theta\cos^2\theta + 2(\pi - \theta)\sin\theta\cos^3\theta + (\pi - \theta)^2\cos^4\theta,$$

$$E(\theta) = -9\sin^2\theta\cos^2\theta - 18(\pi - \theta)\sin\theta\cos^3\theta + (\pi - \theta)^2(1 - 2\cos^2\theta - 8\cos^4\theta),$$

$$E(\theta)F'(\theta)$$

$$= 18\sin^5\theta\cos^3\theta + 90(\pi - \theta)\sin^4\theta\cos^4\theta + 144(\pi - \theta)^2\sin^3\theta\cos^5\theta$$

$$\quad - 2(\pi - \theta)^2\sin^3\theta\cos\theta(1 - 2\cos^2\theta - 8\cos^4\theta) + 72(\pi - \theta)^3\sin^2\theta\cos^6\theta$$

$$\quad - 6(\pi - \theta)^3\sin^2\theta\cos^2\theta(1 - 2\cos^2\theta - 8\cos^4\theta) - 4(\pi - \theta)^4\sin\theta\cos^3\theta(1 - 2\cos^2\theta - 8\cos^4\theta),$$

$$E'(\theta)F(\theta)$$

$$= 18\sin^5\theta\cos^3\theta + 90(\pi - \theta)\sin^4\theta\cos^4\theta + 158(\pi - \theta)^2\sin^3\theta\cos^5\theta + 118(\pi - \theta)^3\sin^2\theta\cos^6\theta$$

$$\quad - 2(\pi - \theta)\sin^6\theta\cos^2\theta - 4(\pi - \theta)^2\sin^5\theta\cos^3\theta - 2(\pi - \theta)^3\sin^4\theta\cos^4\theta$$

$$\quad + 4(\pi - \theta)^2\sin^3\theta\cos^3\theta + 8(\pi - \theta)^3\sin^2\theta\cos^4\theta + 4(\pi - \theta)^4\sin\theta\cos^5\theta + 32(\pi - \theta)^4\sin\theta\cos^7\theta,$$

$$E(\theta)F'(\theta) - E'(\theta)F(\theta) = 2(\pi - \theta)\sin^6\theta\cos^2\theta - 2(\pi - \theta)^2\sin^7\theta\cos\theta - 6(\pi - \theta)^3\sin^4\theta\cos^2\theta - 4(\pi - \theta)^4\sin^3\theta\cos^3\theta.$$

Then the numerator of $\bar{\mathcal{L}}_\xi'(\theta)$ can be written as

$$2(\pi - \theta)\sin^3\theta\,\Psi_\xi(\theta),$$

where

$$\Psi_\xi(\theta) := (1 + \xi)(-\sin\theta + 2(\pi - \theta)\cos\theta) + \frac{1}{\pi^2}[\sin^3\theta\cos^2\theta - (\pi - \theta)\sin^4\theta\cos\theta$$

$$\quad - 3(\pi - \theta)^2\sin\theta\cos^2\theta - 2(\pi - \theta)^3\cos^3\theta].$$

We note that the denominator of $\bar{\mathcal{L}}_\xi'(\theta)$ is always positive and let $\bar{\mathcal{L}}_\xi'(\theta) = 0$. We can get two obvious roots $\theta = 0$ and $\theta = \pi$. Then we want to show that there is only another root in $(\frac{\pi}{3}, \frac{\pi}{2})$. To show that, we first note

$$\Psi_\xi(\frac{\pi}{2}) = -(1 + \xi) < 0, \quad \Psi_\xi(\frac{\pi}{3}) > 0.$$

Therefore, there is at least one root in $(\frac{\pi}{3}, \frac{\pi}{2})$.

We first show that $\Psi_\xi(\theta) < 0$ when $\theta \in (\frac{\pi}{2}, \pi)$. To show that, we have the following inequalities when $\theta \in (\frac{\pi}{2}, \pi)$.

$$0 < \frac{1}{\pi^2}\sin^3\theta\cos^2\theta < \frac{1}{\pi^2}\sin\theta,$$

$$0 < -\frac{1}{\pi^2}(\pi - \theta)\sin^4\theta\cos\theta < -\frac{1}{\pi^2}(\pi - \theta)\cos\theta,$$

$$-\frac{3}{\pi^2}(\pi - \theta)^2\sin\theta\cos^2\theta < 0,$$

$$0 < -\frac{2}{\pi^2}(\pi - \theta)^3\cos^3\theta < -\frac{2}{\pi^2}\frac{\pi^2}{4}(\pi - \theta)\cos\theta = -\frac{1}{2}(\pi - \theta)\cos\theta.$$

From the above inequalities, we have that when $\theta \in (\frac{\pi}{2}, \pi)$,

$$\Psi_\xi(\theta) < (1 + \xi)(-\sin\theta + 2(\pi - \theta)\cos\theta) + \frac{1}{\pi^2}\sin\theta - (\frac{1}{\pi^2} + \frac{1}{2})(\pi - \theta)\cos\theta$$

$$= -\left(1 + \xi - \frac{1}{\pi^2}\right)\sin\theta + \left(\frac{3}{2} + 2\xi - \frac{1}{\pi^2}\right)(\pi - \theta)\cos\theta$$

$$< 0.$$

We next show that $\Psi_\xi(\theta) > 0$ when $\theta \in (0, \frac{\pi}{3})$. First, we note that $\Psi_\xi(\theta)$ can be decomposed as follows.

$$\Psi_\xi(\theta) = \psi_0(\theta) + \xi\psi_1(\theta),$$

where

$$\psi_0(\theta) := -\sin\theta + 2(\pi - \theta)\cos\theta + \frac{1}{\pi^2}[\sin^3\theta\cos^2\theta - (\pi - \theta)\sin^4\theta\cos\theta - 3(\pi - \theta)^2\sin\theta\cos^2\theta$$
$$- 2(\pi - \theta)^3\cos^3\theta],$$
$$\psi_1(\theta) := -\sin\theta + 2(\pi - \theta)\cos\theta.$$

For $\psi_1(\theta)$, we use the facts that $\sin\theta < \theta$ and $\cos\theta > \frac{1}{2}$ on $(0, \frac{\pi}{3})$ and get

$$\psi_1(\theta) > -\theta + 2(\pi - \theta)\cdot\frac{1}{2} = \pi - 2\theta > \frac{\pi}{3} > 0.$$

For $\psi_0(\theta)$, by using Taylor bounds on $(0, \frac{\pi}{3})$

$$\sin\theta > \theta - \frac{\theta^3}{6},\ \sin\theta < \theta,\ \cos\theta > 1 - \frac{\theta^2}{2},\ \cos\theta < 1 - \frac{\theta^2}{2} + \frac{\theta^4}{24},$$

we get

$$\psi_0(\theta) > \theta^2 R(\theta),$$

where

$$R(\theta) := 2\pi - 5\theta + \left(-\frac{7\pi}{4} + \frac{2}{\pi}\right)\theta^2 + \left(\frac{17}{4} - \frac{1}{2\pi^2}\right)\theta^3 + \left(-\frac{11}{4\pi} + \frac{\pi}{2}\right)\theta^4$$
$$+ \left(-\frac{11}{8} + \frac{13}{12\pi^2}\right)\theta^5 + \left(-\frac{7\pi}{96} + \frac{29}{24\pi}\right)\theta^6 + \left(\frac{41}{192} - \frac{59}{108\pi^2}\right)\theta^7$$
$$+ \left(-\frac{5}{24\pi} + \frac{\pi}{192}\right)\theta^8 + \left(-\frac{1}{64} + \frac{161}{1728\pi^2}\right)\theta^9 + \left(-\frac{\pi}{6912} + \frac{1}{64\pi}\right)\theta^{10}$$
$$+ \left(\frac{1}{2304} - \frac{11}{1728\pi^2}\right)\theta^{11} - \frac{1}{2304\pi}\theta^{12} + \frac{1}{6912\pi^2}\theta^{13}.$$

Note that

$$\left(\frac{1}{2304} - \frac{11}{1728\pi^2}\right)\theta^{11} > -2.21\times10^{-4}\,\theta^{10},\ -\frac{1}{2304\pi}\theta^{12} > -1.516\times10^{-4}\,\theta^{10},$$
$$\left(-\frac{5}{24\pi} + \frac{\pi}{192}\right)\theta^8 > -0.0524\,\theta^7,\ \left(-\frac{1}{64} + \frac{161}{1728\pi^2}\right)\theta^9 > -6.7824\times10^{-3}\,\theta^7,$$
$$\left(-\frac{11}{8} + \frac{13}{12\pi^2}\right)\theta^5 > \left(-\frac{11\pi}{24} + \frac{13}{36\pi}\right)\theta^4,$$

by using $\theta < \frac{\pi}{3}$. Then we can get

$$R(\theta) > 2\pi - 5\theta + \left(-\frac{7\pi}{4} + \frac{2}{\pi}\right)\theta^2 + \left(\frac{17}{4} - \frac{1}{2\pi^2}\right)\theta^3 + \left(-\frac{43}{18\pi} + \frac{\pi}{24}\right)\theta^4$$
$$+ \left(-\frac{7\pi}{96} + \frac{29}{24\pi}\right)\theta^6 + 0.099\theta^7 + 4.14\times10^{-3}\theta^{10} + \frac{1}{6912\pi^2}\theta^{13}.$$

After using $\theta < \frac{\pi}{3}$ for $\theta^4$-term, we can further get

$$R(\theta) > 2\pi - 5\theta + \left(-\frac{7\pi}{4} + \frac{2}{\pi}\right)\theta^2 + \left(\frac{373}{108} - \frac{1}{2\pi^2} + \frac{\pi^2}{72}\right)\theta^3$$
$$+ 0.1555\theta^6 + 0.099\theta^7 + 4.14\times10^{-3}\theta^{10} + \frac{1}{6912\pi^2}\theta^{13}$$
$$> 2\pi - 5\theta - 4.8612\theta^2 + 3.54\theta^3 + 0.1555\theta^6 + 0.099\theta^7 + 4.14\times10^{-3}\theta^{10} + \frac{1}{6912\pi^2}\theta^{13}.$$

We note that the coefficients for $\theta^{10}$ and $\theta^{13}$ are positive, so we can focus on the sum of other terms and denote it by $S(\theta)$,

$$S(\theta) := 2\pi - 5\theta - 4.8612\theta^2 + 3.54\theta^3 + 0.1555\theta^6 + 0.099\theta^7.$$

We take the derivative of $S(\theta)$ and use $\theta < \frac{\pi}{3}$ to get

$$S'(\theta) = -5 - 9.7224\theta + 10.62\theta^2 + 0.933\theta^5 + 0.693\theta^6$$

$$< -5 - 9.7224\theta + \theta^2(10.62 + 0.933\frac{\pi^3}{27} + 0.693\frac{\pi^4}{81})$$

$$< -5 - 9.7224\theta + 12.525\theta^2.$$

For the quadratic function $y(\theta) := -5 - 9.7224\theta + 12.525\theta^2$, we can easily check that $y(\theta) < 0$ on $(0, \frac{\pi}{3})$. Hence, $S'(\theta) < 0$ on $(0, \frac{\pi}{3})$ and further

$$S(\theta) > S(\frac{\pi}{3}) > 0.$$

Therefore, we can conclude that $R(\theta) > 0$ and $\psi_0(\theta) > 0$ on $(0, \frac{\pi}{3})$. We have shown that $\Psi_\xi(\theta) > 0$ on $(0, \frac{\pi}{3})$ as desired.

Finally, we want to show that there is only one root in $(\frac{\pi}{3}, \frac{\pi}{2})$. To show that, we find the derivative of $\Psi_\xi(\theta)$.

$$\Psi'_\xi(\theta) = (1 + \xi)(-3\cos\theta - 2(\pi - \theta)\sin\theta)$$

$$+ \frac{1}{\pi^2}[3\sin^2\theta\cos^3\theta - \sin^4\theta\cos\theta - 4(\pi - \theta)\sin^3\theta\cos^2\theta + (\pi - \theta)\sin^5\theta + 6(\pi - \theta)\sin\theta\cos^2\theta$$

$$+ 3(\pi - \theta)^2\cos^3\theta + 6(\pi - \theta)^2\sin^2\theta\cos\theta + 6(\pi - \theta)^3\sin\theta\cos^2\theta].$$

Notice that

$$3\sin^2\theta\cos^3\theta - \sin^4\theta\cos\theta = \sin^2\theta\cos\theta(3\cos^2\theta - \sin^2\theta)$$

$$< \sin^2\theta\cos\theta(3\frac{9}{16}\sin^2\theta - \sin^2\theta)$$

$$= \frac{11}{16}\sin^4\theta\cos\theta < \frac{11}{16}\cos\theta,$$

where the first inequality comes from the fact that $\cos\theta < \frac{3}{4}\sin\theta$ in $(\frac{\pi}{3}, \frac{\pi}{2})$. Furthermore, the following inequalities hold

$$-4(\pi - \theta)\sin^3\theta\cos^2\theta + (\pi - \theta)\sin^5\theta = (\pi - \theta)\sin^3\theta(-4\cos^2\theta + \sin^2\theta)$$

$$< (\pi - \theta)\sin^5\theta < (\pi - \theta)\sin\theta,$$

$$3(\pi - \theta)^2\cos^3\theta + 6(\pi - \theta)^2\sin^2\theta\cos\theta = 3(\pi - \theta)^2\cos\theta(\cos^2\theta + 2\sin^2\theta)$$

$$= 3(\pi - \theta)^2\cos\theta(1 + \sin^2\theta)$$

$$< 6(\frac{2\pi}{3})^2\cos\theta = \frac{8}{3}\pi^2\cos\theta,$$

$$6(\pi - \theta)\sin\theta\cos^2\theta + 6(\pi - \theta)^3\sin\theta\cos^2\theta = 6(\pi - \theta)\sin\theta\cos^2\theta(1 + (\pi - \theta)^2)$$

$$< 6(\pi - \theta)\sin\theta\cos^2\theta(1 + \frac{4}{9}\pi^2)$$

$$< 6(\pi - \theta)\sin\theta\frac{1}{4}(1 + \frac{4}{9}\pi^2)$$

$$= (\pi - \theta)\sin\theta(\frac{3}{2} + \frac{2}{3}\pi^2).$$

As a result, we have

$$\Psi'_\xi(\theta) < -3\cos\theta - 2(\pi - \theta)\sin\theta + \frac{11}{16\pi^2}\cos\theta + \frac{1}{\pi^2}(\pi - \theta)\sin\theta + \frac{8}{3}\cos\theta + (\frac{3}{2\pi^2} + \frac{2}{3})(\pi - \theta)\sin\theta$$

$$= -\left(\frac{1}{3} - \frac{11}{16\pi^2}\right)\cos\theta - \left(\frac{4}{3} - \frac{5}{2\pi^2}\right)(\pi - \theta)\sin\theta$$

$$< 0,$$

which means that $\Psi_\xi(\theta)$ is decreasing in $(\frac{\pi}{3}, \frac{\pi}{2})$. Hence, there is only one root in $(\frac{\pi}{3}, \frac{\pi}{2})$ and we can conclude that it is the global minimizer for any $\xi > 0$. □