# OpenReview forum: "Feature Learning for the High Dimensional Stationary Sch\"odinger Equation with Deep Ritz Method"
_ICLR.cc/2026/Conference — Submitted to ICLR 2026_

### Official Review · Reviewer_GANi · 2025-10-27

**Soundness:** 3
**Presentation:** 2
**Contribution:** 2
**Rating:** 4
**Confidence:** 3

**Summary:**

This paper theoretically analyzes feature learning in simple neural networks when solving high-dimensional stationary Schrödinger equations via the Deep Ritz method. For a single-index (single-neuron) model, they derive convergence rates for Riemannian gradient descent to an approximate global minimum in a general agnostic setting. In the realizable setting, they prove that the minimum Ritz energy is attained when the feature aligns with the feature of the source term. For a two-neuron model (with one feature fixed), the study reveals that the emergence of a new, distinct feature depends critically on the $\ell_2$-regularization strategy: distinct features emerge when only penalizing the outer-layer weight of the second neuron, whereas strong penalization of both outer weights leads to feature collapse. Finally, numerical experiments are provided to validate the results on the loss landscapes and phase transitions for the two-neuron case.

**Strengths:**

- The paper tackles a gap in the theory of deep learning for PDEs. Previous work either only covers generalization & approximation results, or optimization in the "lazy" (NTK) or infinite-width (mean-field) regimes, which often fail to capture how finite-width networks learn relevant features.
- In particular, the authors provide a convergence result in a fully agnostic setting. In the simpler, realizable setting, they are able to derive a clear and interpretable result on how regularization strategies govern feature learning, i.e., encourage/discourage the network from finding new features.
- The theoretical findings for the two-neuron model are well-supported by numerical experiments, validating the predicted loss landscapes, phase transitions, and locations of the minimizers

**Weaknesses:**

- The analysis relies on very simple models (one/two-neuron networks) with a squared ReLU activation. While such a simplification is necessary for a tractable theoretical analysis, it is very far from the architectures used in practice. Moreover, the results do not cover other differentiable activation functions, such as tanh, sigmoid, and the more commonly used GeLU, ELU, and SiLU activations.
- The related works on neural PDE solvers seem to be missing a series of stochastic/SDE-based neural PDE solvers for elliptic equations; see https://arxiv.org/pdf/2112.03749, https://arxiv.org/abs/2001.06145, https://arxiv.org/pdf/2406.03494.
- It should be clearly distinguished which results are derived from (or minor adaptations of) existing results for single/multi-index models (as in Wu (2022); Awasthi et al. (2023); Wang et al. (2023); Gollakota et al. (2023); Zarifis et al. (2024); Diakonikolas et al. (2024)).
- The results only hold for a single PDE (stationary Schrödinger equation) with specific (Neumann) boundary conditions, since the analysis relies on the variational (energy minimization) structure of this elliptic problem. In particular, it is unclear which findings would generalize to other types of PDEs or methods (e.g., PINNs or SDE-based).
- The two-neuron analysis hinges on the assumption that the first feature is *already fixed* to be perfectly aligned with the source feature. This sidesteps the much harder question of the *optimization dynamics* that would lead to this state. Thus, the analysis is purely of the landscape *given* this optimal first feature, not of the full training process to find both features simultaneously.
- The guarantees in the agnostic case only provide a converge results w.r.t. the *best possible loss* achievable by the single-index class. While one could employ approximation results for neural network hypothesis classes, a single-index model is typically a very poor approximation for practically relevant PDEs, making the convergence guarantee practically weak.

Minor: Some readers might be unfamiliar with the term “single- or multi-index models” and it should be explained in the introduction.

**Questions:**

See weaknesses above

---

> ### Author Response · Authors · 2025-11-24
>
> We would first like to thank Reviewer GANi for their time and effort in reviewing our work.
>
> **Answer to Weakness 1:** Thanks for the nice comment on the activation function. We choose to take ReLU-like activation functions because they are widely used in both practice and theoretical analysis of neural networks. However, we can not use ReLU itself since our deep Ritz loss involves integrating the gradient of neural networks. In order to run gradient descent algorithm on the loss, we need the loss to be at least Lipschitz. Squared ReLU is thus a natural choice within the ReLU-like family that enables the Lipschitz continuity of the loss.  It has been shown  that shallow neural networks with squared ReLU or high-order power of ReLU activation enjoy quantitative approximation rate in Sobolev spaces (see e.g. Refs [1] and [2]), making them again natural choices for PDE problems.
>
>
> We expect the landscape and optimization dynamics of GD in the single-neuron case vary a lot when the activation function is different (see a detailed discussion about the effects of activation function and data distribution in the regression problems in Ref [3]).  To better understand the landscape associated to other activations in the two-neuron case,  we have performed new numerical experiments for sigmoid and GELU networks with $d=2$ and observe that they have different angle landscapes. For sigmoid activation, the unique global minimum is achieved at $\theta=\pi$ when the regularization is applied to $\boldsymbol{a}$ and $a_2$ solely. For GELU, when the regularization is applied to $\boldsymbol{a}$, there are two local minimizers at $\theta=0$ and $\theta=\pi$ and the global minimizer changes from $\theta=\pi$ to $\theta=0$ as $\lambda$ increases. When the regularization is applied to $a_2$ solely, there are two local minimizers, one is in $(1,1.5)$ and the other is at $\theta=\pi$ and the global minimizer is at $\theta=\pi$. We will add and  discuss the new numerical results in the revised version.
>
> Ref [1]. Machine Learning For Elliptic PDEs: Fast Rate Generalization Bound, Neural Scaling Law and Minimax Optimality, Yiping Lu, Haoxuan Chen, Jianfeng Lu, Lexing Ying, Jose Blanchet, ICLR 2022.
>
> Ref [2] Approximation Rates for Shallow ReLU Neural Networks on Sobolev Spaces via the Radon Transform, Tong Mao, Jonathan W. Siegel and Jinchao Xu, arXiv:2408.10996.
>
> Ref [3] Learning a Single Neuron with Gradient Methods, Gilad Yehudai and Ohad Shamir, COLT 2020.
>
>
> **Answer to Weakness 2:** Thanks for suggesting the references.  We will add them in the revised version.
>
> **Answer to Weakness 3:** Thanks for bringing this up. We will discuss in detail how our results distinguish from the existing literature. A quick summary is as follows. Wu (2022) studies the learning of single neuron in the realizable setting. Other mentioned works  studied the agnostic GD learning of a single neuron in the regression setting, except that Diakonikolas et al (2024) considers agnostic learning multi-index models using non-GD methods. Our proof borrows  the tool of alignment sharpness from Wang et al (2023) and Zarifs et al. (2024). However, we need to deal with additional technicalities due to the Deep Ritz loss, such as the nonpositivity and  the gradient term. Moreover, since we run the Riemannian gradient descent on the sphere whereas these works run the gradient descent in the Euclidean space, we also need to deal with the error due to  projected gradient.
>
> **Answer to Weakness 4:** As we mentioned earlier, the focus of the paper is to understand the feature learning in a simple problem setting. We stick to homogeneous Neumann boundary condition to avoid dealing with boundary term in the loss. However, the  variational (Ritz) formulation is not essential. We do expect our argument for the single neuron case  still works for PINN losses, even though the boundary loss would bring additional technicalities. We will report the results in future works.  On the other hand, it is indeed unclear whether the proof extends to SDE-based methods as the loss function admits complete different structures.

---

> ### Author Response · Authors · 2025-11-24
>
> **Answer to Weakness 5:** This is a good question. Indeed, it is a much harder question to prove the convergence of the optimization dynamics in the two-neuron setting. In fact, as we mentioned to another reviewer, we are not aware of any convergence guarantee of GD for agnostic learning multiple neurons even in regression problems. Therefore we focus on the loss landscape of the two-neurons in the paper.   Setting the first feature aligned with the source feature is mainly motivated by the optimality result for a single-neuron  shown in Proposition 2.3.  We plan to investigate the training process for both features  in the future.
>
> **Answer to Weakness 6:**
> We agree with the reviewer that a single-neuron class would not be able to well approximate the exact solution of the PDE. However, the goal of the paper is not to obtain optimization guarantee for general neural networks used in the practical training of PDEs, which is a highly challenging theoretical problem and seems to be out of reach for now. Rather, this paper focuses on understanding the mechanism of  feature learning in the simple hypothesis class --- neural networks with with one or two neurons. Even for regression problems, such a feature learning mechanism for a single-neuron was not well understood in the agnostic setting until very recently (see  Wang et al. (2023), Zarifis et al. (2024), Gollakota et al. (2023)). The feature learning analysis becomes extremely difficult in more complex models, including two-layer neural networks or multi-index models even for regression problems (see Diakonikolas et al. (2024)). In fact, we are not aware of any result that proves convergence of GD in the feature learning regime  for more than one neuron in the agonistic setting. Our result (despite focusing on a single hypothesis class) provides the first rigorous understanding of feature learning of neural networks for PDEs. Extending the results to more general networks is of practical importance and challenging and is a natural future direction.

---

### Official Review · Reviewer_yZsG · 2025-10-31

**Soundness:** 3
**Presentation:** 3
**Contribution:** 3
**Rating:** 6
**Confidence:** 3

**Summary:**

The paper investigates feature learning in small neural networks trained with the deep Ritz method (DRM) to solve a stationary Schr\"odinger-type PDE on the unit ball with Neumann boundary conditions. DRM minimizes an energy whose gradient depends only on the known source $f$, not the unknown solution, and the authors study two tiny hypothesis classes: a single-index model $u(x)=\sigma^2(w\cdot x)$ and a two-neuron model $u(x)=a_1\sigma^2(w_1\cdot x)+a_2\sigma^2(w_2\cdot x)$ with squared-ReLU. Directions live on the sphere and are trained by simple Riemannian gradient descent. \\
Result 1: even if the single-index class is misspecified, this training converges quickly to nearly the best value achievable within that class (fast, finite-width optimization). \\
Result 2: if the source is single-index $f(x)=\sigma^2(w^\star\cdot x)$, the learned feature aligns with $w^\star$ (identifiability). \\
Result 3: in the two-neuron case, a high-dimensional analysis reduces the regularized DRM objective to a landscape in the angle $\theta$ between $w_2$ and $w^\star$: joint ridge on $(a_1,a_2)$ induces feature collapse ($\theta=0$) beyond a threshold, whereas penalizing only $a_2$ always yields a nonzero minimizer, so a genuinely new feature emerges. Overall, the work shows that DRM can learn and control finite-width features, with regularization acting as a knob between reuse and diversity.

**Strengths:**

A key strength is that the paper tackles finite-width feature learning head-on: by working with tiny models and the deep Ritz objective, it avoids the usual infinite-width or NTK linearization and still proves concrete optimization guarantees, identifiability under a single-index source, and a crisp, angle-based account of when a second feature emerges or collapses. The energy-based training is appealing in practice because gradients depend only on the known forcing, and the use of Riemannian gradient descent on the sphere keeps the algorithm simple and transparent. The analysis leverage of squared-ReLU and closed-form angular kernels is elegant, giving an interpretable one-dimensional landscape in high dimension and a clean picture of how regularization choice acts as a knob between reuse and diversity; the theory is paired with numerics that qualitatively match the predicted regimes.

**Weaknesses:**

In my opinion the weakness of this approach is that the scope is quite narrow: the PDE is a specific Schr\"odinger-type model on the unit ball with Neumann boundaries, activations are fixed to squared-ReLU, sampling is uniform, and most sharp statements are for one or two neurons with some results relying on a high-dimensional limit, so it is unclear how thresholds and guarantees translate to more realistic geometries, boundary conditions, activations, or wider networks. The identifiability result leans on a single-index structure in the source, which is strong and may fail in multi-directional or noisy settings; the convergence bound in the agnostic case is constant-factor to the best-in-class rather than globally optimal, and practical performance will still depend on step-size, sample budget, and regularization tuning. Finally, while the paper clarifies the mechanisms inside DRM, it offers limited empirical breadth and lacks head-to-head comparisons against alternative PDE training objectives (e.g., residual-based PINNs) on more challenging benchmarks, so the external validity and scalability of the insights remain to be established.

**Questions:**

1. Which activation characteristics—such as polynomial degree, Cho-Saul integral moment structure, and smoothness at 0—are absolutely required for the two-neuron phase behavior and identifiability to hold? Do smooth sigmoids, GELU, or ReLU maintain the same angle landscape?

2. Can you quantify (theoretically or empirically) when the deep Ritz objective is better/worse than residual-based PINNs for the same PDE and model size?

3. How are the optimization landscapes and conditioning different for matched architectures and sampling, and in which regimes does DRM avoid collapse or produce better feature alignment than residual minimization, as demonstrated (or empirically, in controlled tests)?

---

> ### Author Response · Authors · 2025-11-24
>
> We would first like to thank Reviewer yZsG for their time and effort in reviewing our work.
>
> **Answer to Question 1:**
> We choose to take ReLU-like activation functions because they are widely used in practice.
> However, we can not use ReLU itself since our deep Ritz loss involves integrating the gradient of neural networks. In order to run gradient descent algorithm on the loss, we need the loss to be at least Lipschitz. Squared ReLU is thus a natural choice within the ReLU-like family that enables the Lipschitz continuity of the loss. Second, it has been shown  that shallow neural networks with squared ReLU or high-order power of ReLU activation enjoy quantitative approximation rate in Sobolev spaces (see e.g. Refs [1] and [2]), making them again natural choices for PDE problems. We will elaborate in detail on the choice of activation function in the revised version.
>
> Ref [1]. Machine Learning For Elliptic PDEs: Fast Rate Generalization Bound, Neural Scaling Law and Minimax Optimality, Yiping Lu, Haoxuan Chen, Jianfeng Lu, Lexing Ying, Jose Blanchet, ICLR 2022.
>
> Ref [2] Approximation Rates for Shallow ReLU Neural Networks on Sobolev Spaces via the Radon Transform, Tong Mao, Jonathan W. Siegel and Jinchao Xu, arXiv:2408.10996.
>
> For other activation factions, we have performed new numerical experiments on the  landscapes for sigmoid and GELU networks with $d=2$ and observe that they have different angle landscapes. For sigmoid activation, the unique global minimum is achieved at $\theta=\pi$ when the regularization is applied to $\boldsymbol{a}$ and $a_2$ solely. For GELU, when the regularization is applied to $\boldsymbol{a}$, there are two local minimizers at $\theta=0$ and $\theta=\pi$ and the global minimizer changes from $\theta=\pi$ to $\theta=0$ as $\lambda$ increases. When the regularization is applied to $a_2$ solely, there are two local minimizers, one is in $(1,1.5)$ and the other is at $\theta=\pi$ and the global minimizer is at $\theta=\pi$. We will add and  discuss the new numerical results in the revised version.
>
> **Answer to Question 2:**
> Comparing the performance of deep Ritz method with PINNs is an interesting question, but is not the focus of this paper. Our focus is to understand the feature learning mechanism of deep Ritz method in simple settings. Extending the analysis to residual-based methods would be an interesting future direction.  We would like to refer the reviewer to the following papers for a detailed theoretical comparison of the two methods.
>
> Ref [1]. Machine Learning For Elliptic PDEs: Fast Rate Generalization Bound, Neural Scaling Law and Minimax Optimality, Yiping Lu, Haoxuan Chen, Jianfeng Lu, Lexing Ying, Jose Blanchet, ICLR 2022.
>
> Ref [2]. Refined Generalization Analysis of the Deep Ritz Method and Physics-Informed
> Neural Networks, Xianliang Xu, Ye Li and Zhongyi Huang, ICML 2025.
>
> **Answer to Question 3:** We appreciate it if the reviewer can clarify what the terms "conditioning" and "matched architectures" refer to. Considering the comparison of performance for feature alignment between DRM and residual minimization, as we mentioned earlier, it is an interesting question, but is not the focus of this paper. We expect our proof techniques will be still effective for analyzing the feature learning of residual-based methods, but a rigorous characterization of the  feature learning mechanism will be reported in future work.

---

### Official Review · Reviewer_3ZWX · 2025-11-06

**Soundness:** 3
**Presentation:** 3
**Contribution:** 2
**Rating:** 4
**Confidence:** 4

**Summary:**

This paper studies feature learning for PDEs within the framework of the deep Ritz method applied to the stationary Schrödinger equation. By restricting the hypothesis space to a single-index model, the authors prove that Riemannian gradient descent converges to a loss within  $\epsilon$ of a constant multiple of the optimal loss. They also analyze the minimizers of the energy functional under two models, assuming the source term is itself a single-index function.

**Strengths:**

- The paper focuses on the optimization aspect of neural PDE solvers, a fundamental but comparatively less understood topic.
- To my knowledge, using single-index models to study neural PDE solvers is novel in the literature.
- The presentation is well organized, and the writing is clear.

**Weaknesses:**

Overall, the paper analyzes optimization in rather restrictive settings. These assumptions seem chosen primarily to make the analysis feasible, but they limit the relevance of the results to practical PDE solving. If I could rate on a continuous scale, my score would be a 3, between the criteria for 2 and 4.

Specific concerns:
- The link function is restricted to squared ReLU, with no justification. This is a strong and arguably artificial assumption. Additionally, the proof appears challenging to extend directly to other link functions.

- The convergence result only guarantees reaching a minimum within a factor $\gamma>1$ of the global minimum. But within such a narrow hypothesis space, the global minimum itself may be large. In that case, the result is of limited significance.

- Boundary conditions are essential in PDEs, yet they are not incorporated into the algorithmic framework at all.

**Questions:**

- Related to the second weakness: do the authors have any approximation results showing that the global minimum in the single-index setting is actually small? This would help justify the meaningfulness of the convergence guarantee.

- Can anything be said if a boundary penalty is added to the loss? Alternatively, are there settings in which the hypothesis class inherently satisfies the boundary conditions?

- In Section 2.3, the assumption $w_1 = w^*$ seems to imply a staged training process: first learn one index fully, then optimize the second. If all parameters are trained jointly (as in practice), can the authors comment anything on the resulting landscape?

- The numerical experiments rely on simplifications that $d \to \infty $. Can the authors provide experiments for the original finite-dimensional problem to give any insight into practical relevance?

---

> ### Author Response · Authors · 2025-11-24
>
> We would like to thank Reviewer 3ZWX for their time and effort in reviewing our work.
>
> **Answer to Weakness 1:** We consider the squared ReLU activation for at least two reasons.
> First, ReLU activation is perhaps most studied in the feature learning literature of neural networks for regression problems. Here we aim to extend the understanding of the feature learning from regression to PDEs. However, we can not use ReLU itself since our deep Ritz loss involves integrating the gradient of neural networks. In order to run gradient descent algorithm on the loss, we need the loss to be at least Lipschitz. Squared ReLU is thus a natural choice within the ReLU-like family that enables the Lipschitz continuity of the loss. Second, it has been shown  that shallow neural networks with squared ReLU or high-order power of ReLU activation enjoy quantitative approximation rate in Sobolev spaces (see e.g. Refs [1] and [2]), making them again natural choices for PDE problems. We will elaborate in detail on the choice of activation function in the revised version.
>
> Ref [1]. Machine Learning For Elliptic PDEs: Fast Rate Generalization Bound, Neural Scaling Law and Minimax Optimality, Yiping Lu, Haoxuan Chen, Jianfeng Lu, Lexing Ying, Jose Blanchet, ICLR 2022.
>
> Ref [2] Approximation Rates for Shallow ReLU Neural Networks on Sobolev Spaces via the Radon Transform, Tong Mao, Jonathan W. Siegel and Jinchao Xu, arXiv:2408.10996.
>
> **Answer to Weakness 2 \& Question 1:** We agree with the reviewer that the global minimum of the loss in the single-neuron class may not be small. However, the goal of the paper is not to obtain optimization guarantee for general neural networks used in the practical training of PDEs, which is a highly challenging theoretical problem and seems to be out of reach for now. Rather, this paper focuses on understanding the mechanism of  feature learning in the simple hypothesis class --- neural networks with one or two neurons. Even for regression problems, such a feature learning mechanism for a single-neuron was not well understood in the agnostic setting until very recently (see  Awasthi et al. (2023), Wang et al. (2023), Zarifis et al. (2024), Gollakota et al. (2023)). The feature learning analysis becomes extremely difficult in more complex models, including two-layer neural networks or multi-index models even for regression problems (see Diakonikolas et al. (2024)). In fact, we are not aware of any result that proves convergence of GD in the feature learning regime  for more than one neuron in the agonistic setting. Our result (despite focusing on a single hypothesis class) provides the first rigorous understanding of feature learning of neural networks for PDEs.  Extending the results to more general networks is of practical importance and challenging and will be investigated in future work.
>
> **Answer to Weakness 3 \& Question 2:** We agree that boundary conditions are essential in PDEs. Here our deep Ritz loss incorporates the natural (or homogeneous Neumann) boundary condition. Either adding nonhomogeneous Neumann boundary condition or other boundary conditions (Dirichlet or Robin) will significantly complicate the analysis, since the features from both the source and nonhomogeneous boundary data need to be learned.     We decide to focus on learning  the interior features only to avoid additional technicalities due to multiple feature sources. Extending the analysis to taking into account  other boundary conditions will be a natural future direction.
>
> **Answer to Question 3:** Thanks for the nice question. One major motivation of fixing one feature vector to be $w^\ast$ is that  when the hypothesis function $u$ is defined by a single index model $u(x)=\sigma^2(w\cdot x)$, the minimum value of the Ritz energy function $\mathcal{E}(w)$ is achieved when $w=w^*$ (see Proposition 2.3). In fact, one can also show that if the hypothesis function is $a_1\sigma^2(w_1\cdot x) + a_2 \sigma^2(w_2\cdot x)$ with $a_i >0$, then one of the features also aligns with $w^\ast$. We will add a remark on this in the revised version. However, the landscape for the loss $\mathcal{L}$ defined in equation (7) on both $\boldsymbol{a}$ and $\boldsymbol{w}$ is more delicate. In particular, the optimal features may not align with that of the source feature $w^\ast$.
>
> **Answer to Question 4:** We will plot the loss function for the original finite-dimensional problem and include the graphs in the revised version.

---

> > ### Comment · Reviewer_3ZWX · 2025-11-26
> >
> > Thanks to the authors for the clarifications in their responses. I would like to maintain my score.

---

### Meta-Review · Area_Chair_onWW · 2025-12-28

**Summary:**

This paper studies the feature learning problem of the deep Ritz method for the static Schrodinger equation with Neumann boundary condition. There are two main contributions of the paper: analyze the convergence for a Riemannian gradient descent in an agnostic setting and examine the training loss landscape when the source term is a single- or multi-index model.

The reviewers raised several common weakness and questions:

- the link function is ReLU^2, which is restrictive and lack of justification;
- boundary condition is not incorporated into the convergence analysis;
- two-neuron analysis (multi-index model) relies on the aligned first feature is too strong;
- convergence bound in the agnostic case is within constant factor of the optimal in class.

**Reviewer Concerns:**

In response, the authors clarified the motivation of using ReLU^2 activation as a popular choice for regression problems and the necessity for running gradient method on the training loss. Additional numeric experiments were tested on other activations.

While finding the rebuttal partially addressed the reviewers’ concerns and feature learning for PDE problems is challenging, I still view the current setting is a bit contrived and share the concern for lack of boundary analysis to the feature learning in the PDE setting.

**Reviewer Scores:**

3 reviewers submitted their comments and scores (4/6/4) with confidence (3/3/4), with average score 4.67 and average confidence 3.33.  One reviewer mentioned to maintain score (score: 4 & confidence 4).

---

### Decision · Program_Chairs · 2026-01-26

Reject